# Organize the Web: Constructing Domains Enhances Pre-Training Data Curation

**Alexander Wettig** [1 2]   **Kyle Lo** [2]   **Sewon Min** [3 2]   **Hannaneh Hajishirzi** [2 4]   **Danqi Chen** [1]   **Luca Soldaini** [2]

## Abstract

Modern language models are trained on large, unstructured datasets consisting of trillions of tokens and obtained by crawling the web. The unstructured nature makes it difficult to reason about their contents and develop systematic approaches to data curation. In this paper, we unpack monolithic web corpora by developing taxonomies of their contents and organizing them into domains. We introduce **WebOrganizer**, a framework for organizing web pages in terms of both their topic and format. Using these two complementary notions of domains, we automatically annotate pre-training data by distilling annotations from a large language model into efficient classifiers. This allows us to study how data from different domains should be mixed to improve models on downstream tasks, and we show that we can combine insights about effective topics and formats to further boost performance. We demonstrate that our domain mixing also improves existing methods that select data based on quality. Furthermore, we study and compare how quality-based methods will implicitly change the domain mixture. Overall, our work demonstrates that constructing and mixing domains provides a valuable complement to quality-based data curation methods, opening new avenues for effective and insightful pre-training data curation.

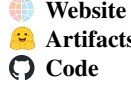

| | | |
|---|---|---|
| 🌐 | **Website** | weborganizer.allen.ai |
| 🤗 | **Artifacts** | hf.co/WebOrganizer |
| ⌗ | **Code** | CodeCreator/WebOrganizer |

[1]Princeton Language and Intelligence, Princeton University [2]Allen Institute for Artificial Intelligence [3]University of California, Berkeley [4]Paul G. Allen School of Computer Science & Engineering, University of Washington. Correspondence to: Alexander Wettig <awettig@cs.princeton.edu>.

*Proceedings of the 42nd International Conference on Machine Learning*, Vancouver, Canada. PMLR 267, 2025. Copyright 2025 by the author(s).

## 1. Introduction

Curating good training data is crucial for enhancing the capabilities of language models. Early pre-training datasets, like the Pile (Gao et al., 2020) or RedPajama (TogetherAI, 2023), were created by curating data from multiple sources—such as Wikipedia, Reddit, or BookCorpus (Zhu et al., 2015)—giving rise to the research problem of how to balance these domains[2] (Xie et al., 2023a). However, as the demand for data has grown to trillions of tokens, the majority of data is now obtained from crawling the web, and the importance of curating domains has diminished.

Recent efforts in data curation, such as FineWeb (Penedo et al., 2024) and DCLM (Li et al., 2024), produce multi-trillion-token datasets with CommonCrawl as the singular source, offering no summary of their contents. In the absence of domains, the focus has shifted to cleaning corpora using heuristic rules (Raffel et al., 2020; Rae et al., 2021; Penedo et al., 2023) and quality filters (Wettig et al., 2024; Sachdeva et al., 2024; Penedo et al., 2024; Li et al., 2024).

In this paper, we propose WebOrganizer, a framework to construct meaningful domains for monolithic web corpora. Our approach consists of designing taxonomies for unstructured web content, and scaling automatic labeling of documents according to these taxonomies by distilling a large language model classifier (Llama-3.1-405B-Instruct) to small and efficient models (140M parameters). WebOrganizer establishes a rich, two-dimensional structure for pre-training data by introducing two complementary domain taxonomies—**topic** and **format**—which classify web pages into 24 categories based on subject matter and style, respectively. This paper, for instance, would fall under the *Science & Technology* topic and the *Academic Writing* format. Figure 1 provides an overview of these domains and demonstrates how WebOrganizer shines a light on the composition of different types of internet content in a cleaned pre-training corpus derived from CommonCrawl. We also compare our domains to $k$-means clustering of document embeddings, and find that the clusters

---

[2]Throughout the paper, we use the term *domain* to denote dataset partitions, rather than conventional web domains, which we will refer to as *URL domains*.

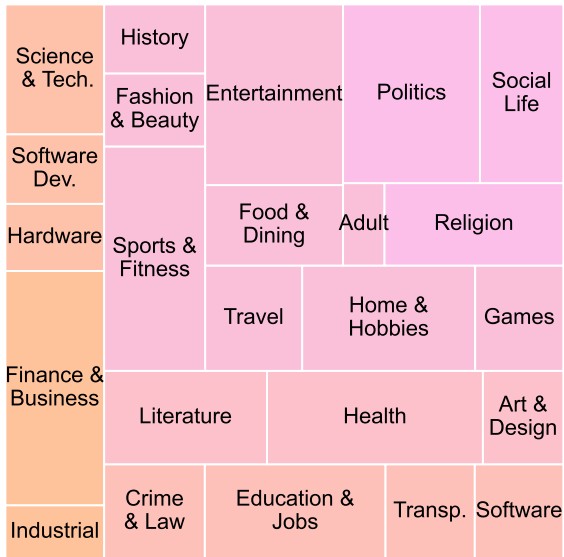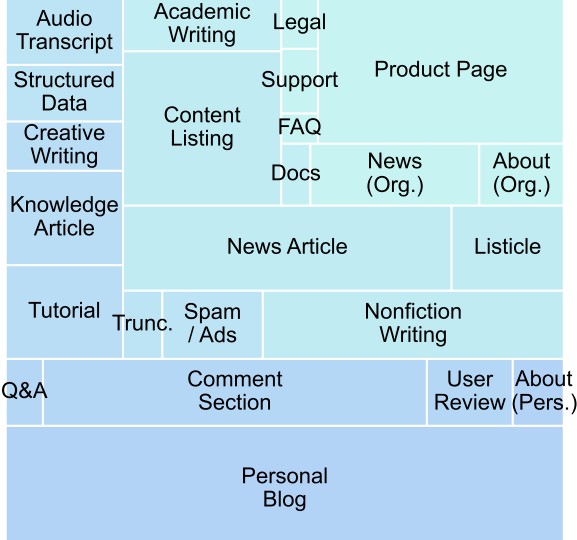

Figure 1: We construct **topic domains** (left) and **format domains** (right) to organize pre-training corpora. The areas visualize the number of tokens per domain in a cleaned pre-training corpus based on CommonCrawl. See Appendix A for detailed definitions of the categories. We provide an interactive explorer of the domains at `weborganizer.allen.ai`.

mostly align with topics and do not reveal different formats.

How effective are these domains for data curation? Partitioning a corpus into domains provides rich affordances for data curation, as we can flexibly up or down-sample domains. More importantly, it enables principled methods that can explore possible data mixtures in systematic ways and optimize the domain proportions to meet the objectives of data curation. We adapt the RegMix framework (Liu et al., 2024) to predict which domains should be up-sampled to improve two downstream tasks, MMLU and HellaSwag, which are commonly used for measuring the quality of pre-training data. For example, we find that up-sampling documents from the *Science & Technology* topic favors MMLU while the *Tutorial* format suits HellaSwag.

Our experiments show that constructing domains and optimizing their mixture towards specific tasks is effective. The reweighted topics, formats, and $k$-means clusters all improve downstream performance across a range of downstream tasks. Furthermore, we show that, since the topic and format domains capture different aspects of web pages, we can combine their data mixtures, which boosts performance considerably—matching the overall results of selecting documents with the FineWeb-Edu quality filter (Penedo et al., 2024). Even more compelling, we find that our optimized domain mixtures work well with quality filters and further enhance their performance. Notably, the average accuracy of FineWeb-Edu increases from 54.2% to 56.2% when adding domain mixing—almost doubling the gain of FineWeb-Edu over a 51.6% baseline accuracy.

Finally, we observe that data selection with quality filters implicitly changes the domain proportions of the dataset, and we quantify how much their performance gain can be accomplished from this domain mixing alone. We observe that the FineWeb-Edu quality filter has similar domain preferences to the mixtures optimized for MMLU and the implicit topic and format mixture retains up to 84% of the performance gains from quality filtering. Meanwhile, while the DCLM-fasttext quality filter (Li et al., 2024) amplifies certain formats, we find that its implicit data mixtures perform considerably worse, suggesting that it utilizes aspects of quality beyond broad domain effects.

We open-source WebOrganizer as a tool for understanding, documenting and curating pre-training data. To encourage future work, we include the code for constructing domains and training domain classifiers, as well as the annotated pre-training corpus.

## 2. Constructing Domains for Web-Scale Data

State-of-the-art language models rely overwhelmingly on *web-crawled* training data (Baack, 2024; Raffel et al., 2020; Brown et al., 2020; Rae et al., 2021; Penedo et al., 2023; Soldaini et al., 2024; Penedo et al., 2024; Li et al., 2024)—with open-source research typically resorting to data provided by the CommonCrawl foundation[3]. Unlike the large-scale ImageNet dataset (Deng et al., 2009), which was collected according to an explicit conceptual hierarchy,

---

[3] `commoncrawl.org`

these web corpora simply contain all web pages adhering to certain filtering rules, resulting in trillions of tokens of text without an inherent structure. While it is common practice to include specially curated domains, such as Wikipedia or StackOverflow (Touvron et al., 2023; TogetherAI, 2023; Soldaini et al., 2024), these additions are comparatively small and do not demystify the vast amount of data within CommonCrawl.

Our practices of data curation are opaque and uninformed without a firm understanding of how these large-scale corpora are internally composed. In this paper, our approach is to *design* domain taxonomies to address this short-coming. We first lay out the desirable properties of such domains, and then we describe our method for creating taxonomies and annotating pre-training datasets, and finally compare our taxonomy-driven domains to a baseline that partitions a corpus via $k$-means clusters.

**Desiderata** Since a corpus can be partitioned in exponentially many ways, we seek domains that produce human insights into pre-training corpora and our domains should align with meaningful human categories. To facilitate human exploration, we also aim for a compact number of domains that capture high-level trends and allow for a concise representation of the corpus. Therefore, each domain should also have a reasonable amount of presence in the corpus. For example, URL domains would be too granular, as there are 18.5k URL domain names with more than 1k documents in a 200B token subset of CommonCrawl and 14.7M URL domain names with fewer documents (see Figure 5 in appendix). A smaller set of domains also decreases the chance of domain conflicts and ambiguities, and makes it easier to learn how to rebalance these domains.

### 2.1. Human-in-the-loop design of domain taxonomies

We design two domain taxonomies for WebOrganizer to capture the **topic** and **format** of web pages, respectively. These are meant to capture complementary characteristics: The topic should describe the subject matter of content, whereas the format concerns its style, intent and venue.[4]

We start by reviewing existing fine-grained web taxonomies, specifically the crowd-sourced curlie.org web directory, Google Adsense, the Wikipedia ontology, and the most frequent URL domains. We identify common themes and propose coarse-grained topic and format definitions, which we iteratively refine by prompting Llama-3.1-405B-Instruct (Dubey et al., 2024) to classify CommonCrawl samples and reviewing these annotations.

Following our desiderata, we consolidate less frequently

occurring topics into topic clusters—for example, our *Industrial* topic spans manufacturing, mining, agriculture, and utilities, mathematics is subsumed in *Science & Technology*; in terms of formats, cooking recipes become part of *Tutorials*. We also adjust the categories to match the abilities of language models and what they can deduce from seeing only the URL and text contents of a web page. For example, we observe that models are uncertain when choosing between comment sections and discussion forums, motivating us to merge these formats. In other instances, we add guidelines for resolving ambiguous cases. We eventually settle on 24 categories per taxonomy (see definitions and prompts in Appendix A).

Our approach of proposing taxonomies in natural language and refining them based on model annotations is flexible and can be used for other purposes—for instance, to annotate a corpus of scientific papers with detailed subject areas, or to taxonomize the data within each of our domains to build a nested hierarchy. Unlike recent techniques for automatically constructing taxonomies with large-language models (Chen et al., 2021a; Mishra et al., 2024; Pham et al., 2024), our approach requires human effort, but also benefits from human oversight and domain expertise.

### 2.2. Training domain classifiers for scaling annotations

While useful during taxonomy development, it would be extremely expensive to annotate a web-scale corpus with a large language model. Therefore, we enable WebOrganizer by fine-tuning small classifier model to imitate the annotations of Llama-3.1-405B-Instruct using a soft knowledge distillation loss (Hinton et al., 2015). We initialize the classifiers with gte-base-en-v1.5 (Li et al., 2023b)—a 140M parameter embedding model with a 8192 token context window—and train them in two stages to improve their coverage over diverse web content. In the first stage, we train with 1M annotations from the cheaper Llama-3.1-8B-Instruct model, followed by 80K high-quality annotations from Llama-3.1-405B-Instruct. In Appendix B, we discuss the setup in more detail and show how the two-stage training is useful for improving the classifier accuracy. We use the topic and a format classifiers to annotate a 200B pre-training corpus, which is based on CommonCrawl and cleaned using heuristic rules (Penedo et al., 2023) and deduplication (Soldaini et al., 2024).

### 2.3. Domain statistics

Figure 1 gives an overview of the topic and format domains provided by WebOrganizer and visualizes their proportions in the pre-training corpus. Figure 2 shows the highest values of the normalized pointwise-mutual information be-

---

[4]This distinction has also been made by van der Wees et al. (2015) in terms of topic and genre.

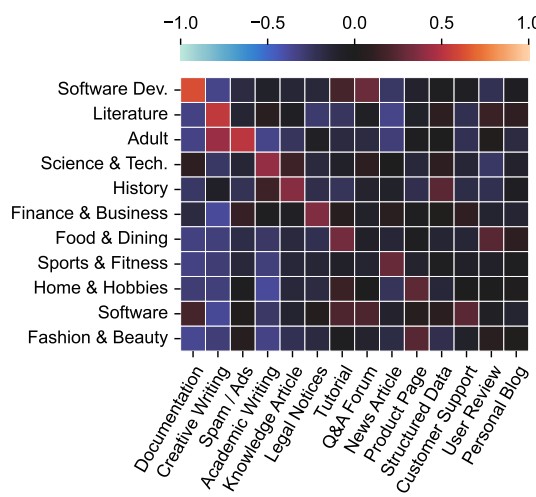

Figure 2: We visualize the 15 highest co-occurrences in the normalized pointwise mutual information (NPMI) matrix between topics (y-axis) and formats (x-axis). Figure 6 shows the full matrix, where most entries are close to zero.

tween topic annotations $T$ and formats $F$,

$$\text{NPMI}(T; F) = \log \frac{p(T, F)}{p(T)p(F)} \bigg/ \log \frac{1}{p(T, F)},$$

where a value of 0 suggests independence and 1 implies complete co-occurrence. The majority of entries are close to zero with reasonable exceptions for pairs such as *Documentation* and *Software Development*. The normalized mutual information measures the overall level of redundancy, $\text{NMI}(T; F) = \frac{2I(T; F)}{H(T)+H(F)} \approx 0.10$, which is close to zero and suggests that an independence assumption approximates the domain product well.

### 2.4. Comparison to $k$-means clustering

Clustering is a natural baseline for partitioning a corpus and has previously been used for training expert models (Gururangan et al., 2023). We use the *gte-base-en-v1.5* model (Li et al., 2023b) to compute document embeddings for the 200B pre-training corpus and run $k$-means using a distributed implementation by Vo et al. (2024). We obtain 24 clusters that are more evenly balanced than our domains, but lack inherent natural language descriptions. Interestingly, we find that the $k$-means cluster assignments $C$ tend to reflect the web site topic more strongly than its format (since $\text{NMI}(C; T) \approx 0.46$ vs. $\text{NMI}(C; F) \approx 0.13$, also see Figure 7 in the appendix). Furthermore, the trend is similar even with more fine-grained $k$-means domains of 576 clusters (NMI statistics remain within $\pm 0.03$). The orthogonal nature of the format domains suggests that careful human-in-the-loop taxonomies can provide richer data annotations than clustering document embeddings alone.

## 3. Optimizing Domain Mixtures for Downstream Tasks

The promise of organizing a corpus with WebOrganizer is that we can learn the importance of each domain in a principled way. In this section, we study how to rebalance these domains to align with the needs of downstream tasks. This reflects the typical goal of data curation, which is to improve task performance when using a dataset for training language models (Wettig et al., 2024; Penedo et al., 2024)—for instance, this is the protocol of the DataComps-LM competition (Li et al., 2024).

**Mixture prediction** While many methods have been developed to optimize domain mixtures (Xie et al., 2023a; Chen et al., 2023; Albalak et al., 2023; Fan et al., 2024; Chen et al., 2024; Jiang et al., 2024), most focus on minimizing the in-distribution loss. We decide to use Reg-Mix (Liu et al., 2024) due to its simplicity and adapt it to optimize the mixture distribution for downstream tasks. For each set of domains—topics, formats, and $k$-mean clusters—we train 512 models of 50M parameters for 1B tokens and fit a gradient-boosted tree regression model (Ke et al., 2017). We make a mixture prediction by searching for the lowest loss in the input space of the regression model, restricting our search to mixtures which upsample domains at most $6.5\times$, which ensures that we do not exhaust all documents when selecting training data in Section 4. We use an iterative search method, deviating from RegMix. Appendix C discusses our implementation in detail.

**Target tasks** Whereas RegMix (Liu et al., 2024) uses the C4 loss as a proxy loss for task performance, we directly focus on two popular question-answering tasks, MMLU (Hendrycks et al., 2021) and HellaSwag (Zellers et al., 2019), as well as their average. MMLU requires diverse world knowledge and problem solving abilities, whereas HellaSwag is an adversarially filtered dataset for common-sense reasoning. To avoid contamination, we use the training and validation set of these two tasks, respectively. We seek a mixture that minimizes the next-token prediction loss over the correct response normalized by the response length (bits-per-byte) given a 5-shot prompt. This loss has also been used for extrapolating model task performance (Bhagia et al., 2024).

**Predicted mixtures** Figure 3 visualizes the training distributions predicted by RegMix across the topic and format domains constructed by WebOrganizer. We observe that the two target tasks call for remarkably different data mixtures. The MMLU mixture heavily upsamples *Science & Technology*, followed by *History* and *Health*, and in terms of formats, promotes *Academic Writing* and *Q&A Forums*.

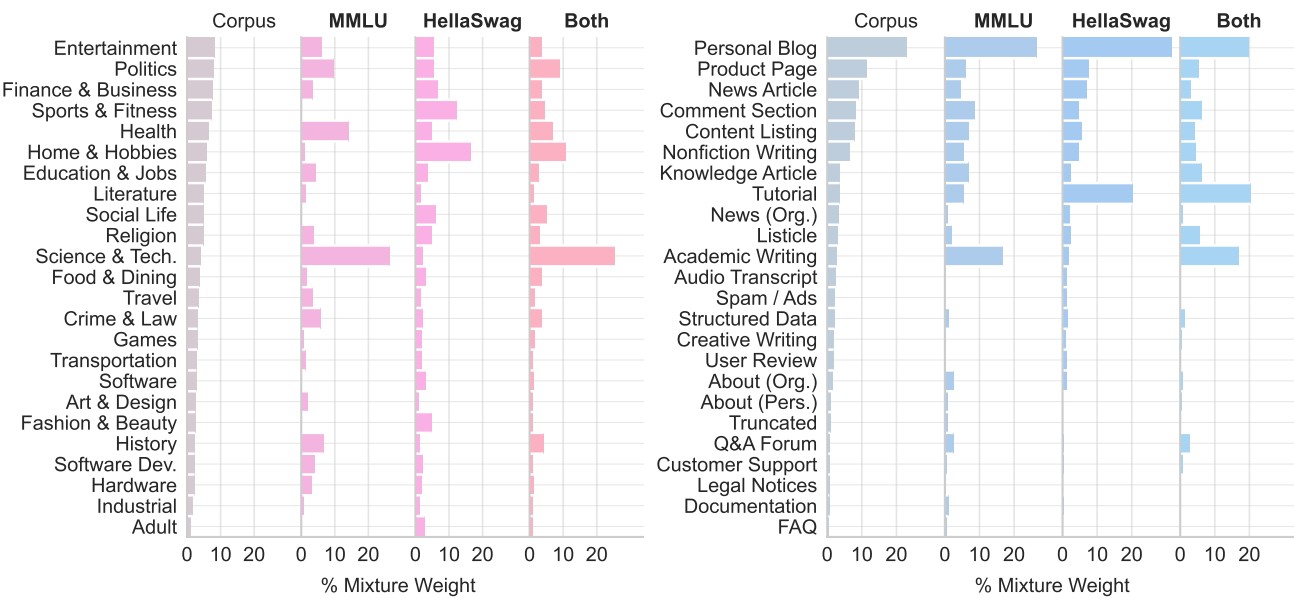

Figure 3: The corpus proportions of our topic domains (left) and formats (right), and the training mixtures predicted by RegMix for targeting MMLU, HellaSwag, and both tasks. Numerical values can be found in Table 9 in the appendix.

HellaSwag exhibits smoother mixtures, notably amplifying *Home & Hobbies* and *Fashion & Beauty*, and strongly boosting *Tutorials*. Meanwhile, the mixtures tailored towards the average of both tasks tend to combine the prominent components of each task mixture. We provide predicted mixtures for additional downstream tasks, including HumanEval (Chen et al., 2021b) and Natural Questions (Kwiatkowski et al., 2019) in Appendix D.

## 4. Evaluating Pre-Training Data Curation with WebOrganizer

We demonstrate the practical value of constructing domains with WebOrganizer by training models with the domain mixtures produced by RegMix in Section 3. We show how to combine data mixing for topics and formats, and how domains can be used together with quality filters.

### 4.1. Experimental Setting

All our experiments are implemented in the DataComps-LM (DCLM) framework (Li et al., 2024), using the 1b-1x competition pool. We follow best practices and use heuristic filters, followed by deduplication to reduce the 1.6T raw token pool to a base corpus of 200B tokens. From this dataset, we select 29B tokens by sampling according to a domain mixture and train a 1B parameter model. Full details of our experimental setup can be found in Appendix E.

**Evaluation suite** We use OLMES (Gu et al., 2024) to evaluate models and their domain mixtures. We use a 5-

shot setting on a suite of 9 tasks: MMLU (Hendrycks et al., 2021), HellaSwag (HSwag) (Zellers et al., 2019), PIQA (Bisk et al., 2020), WinoGrande (WinoG) (Sakaguchi et al., 2021), CommonSenseQA (CSQA) (Talmor et al., 2019), Social IQa (SIQA) (Sap et al., 2019), ARC-easy/challenge (ARC-e/ARC-c) (Clark et al., 2018), and OpenBookQA (OBQA) (Mihaylov et al., 2018). OLMES measures task performance in both the multiple-choice format and a cloze formulation, as well as curating few-shot examples, producing a more reliable evaluation for smaller models.

### 4.2. Topic × Format selection

We construct a new taxonomy, consisting of all 576 pairs of topic and format domains. Finding a training mixture of this cardinality would be expensive and sensitive to noise. Here, we make the assumption that we can select topics and formats independently, and select data according to

$$\tilde{P}_{T \times F}(\text{topic, format}) = \tilde{P}_T(\text{topic})\tilde{P}_F(\text{format}),$$

where $\tilde{P}_T(\text{topic})$ and $\tilde{P}_F(\text{format})$, are the predictions from separate RegMix pipelines. In practice, there are cases where $\tilde{P}_T(\text{topic})\tilde{P}_F(\text{format})$ can exceed the amount of data available for that pair. In such cases, we take all available documents and up-sample everything else to compensate.

### 4.3. Combining quality filters and domain mixing

Quality filters assign scores to individual documents (Xie et al., 2023b; Wettig et al., 2024; Sachdeva et al., 2024), and select data at a more granular level than is possible

Table 1: Evaluating the benefits of domain mixing, where the domain mixtures are tailored towards MMLU and HellaSwag (Figure 3). All models are trained in the `1b-1x` setting from DCLM (Li et al., 2024) The baseline corpus is pre-processed with heuristic filtering and deduplication and forms the basis for the other data curation methods.

| Data Curation | MMLU | HSwag | PIQA | WinoG | CSQA | SIQA | ARC-e | ARC-c | OBQA | Avg |
|---|---|---|---|---|---|---|---|---|---|---|
| Baseline corpus | 30.3 | 57.5 | 71.3 | 56.1 | 59.0 | 49.9 | 62.2 | 34.0 | 44.0 | 51.6 |
| + Clusters | 31.8 | 59.4 | 73.4 | 58.2 | 58.7 | 50.7 | 66.1 | 35.2 | 44.8 | 53.2 |
| + Topic | 31.4 | 56.2 | 72.1 | 54.8 | 61.3 | 47.8 | 70.3 | 40.6 | 49.0 | 53.7 |
| + Format | 31.7 | 60.9 | 74.1 | 56.9 | 60.1 | 47.4 | 65.8 | 35.9 | 47.6 | 53.4 |
| + Topic × Format | 32.7 | 60.1 | 73.4 | 56.5 | 62.3 | 49.3 | 69.7 | 38.8 | 49.0 | 54.6 |
| | ↑2.4 | ↑2.6 | ↑2.1 | ↑0.4 | ↑3.3 | ↓0.6 | ↑7.5 | ↑4.8 | ↑5.0 | ↑3.0 |
| FineWeb-Edu | 34.3 | 56.0 | 69.9 | 57.7 | 60.0 | 47.9 | 71.9 | 42.3 | 48.2 | 54.2 |
| + Topic × Format | 34.2 | 62.5 | 73.3 | 57.1 | 63.0 | 49.4 | 72.2 | 43.3 | 50.8 | 56.2 |
| | ↓0.1 | ↑6.5 | ↑3.4 | ↓0.6 | ↑3.0 | ↑1.5 | ↑0.3 | ↑1.0 | ↑2.6 | ↑2.0 |
| DCLM-fasttext | 33.4 | 59.0 | 70.5 | 58.8 | 63.2 | 50.7 | 71.4 | 39.8 | 48.8 | 55.1 |
| + Topic × Format | 33.8 | 63.1 | 74.3 | 57.6 | 62.7 | 49.8 | 73.4 | 42.2 | 47.8 | 56.1 |
| | ↑0.4 | ↑4.1 | ↑3.8 | ↓1.2 | ↓0.5 | ↓0.9 | ↑2.0 | ↑2.4 | ↓1.0 | ↑1.0 |

with domain rebalancing. Therefore, they are a powerful baseline. We compare to two state-of-the-art quality filters: FineWeb-Edu (Penedo et al., 2024), a 110M parameter model distilled from prompting Llama-3-70B to rate the educational value of web pages, and DCLM-fasttext (Li et al., 2024), a bigram model trained to identify text resembling a reference corpus consisting mostly of GPT-4 conversations. For both methods, we select all the highest-ranking documents until the token budget is reached.

We explore a simple strategy for composing quality filters and domain mixtures: We use the domain mixture to determine the desired number of tokens from each domain subset. Then we perform the data selection with the quality filter separately for each subset—effectively varying the quality threshold per domain, depending on the mixture.

### 4.4. Results

Table 1 shows the results of our main experiments with mixtures optimized for both MMLU and HellaSwag. In the first setting, we consider how domain mixing improves upon the inherent data mixture of the baseline corpus. Then, we show how domain mixing also improves the performance of quality filtering. Results for individual task mixtures are reported in the appendix (Table 10).

**Domain mixing is broadly effective** We observe that reweighting the domain proportions of the pre-training corpus improves downstream performance across all three of the topic, format, and $k$-means cluster domains (rows 1-4 in Table 1). Rebalancing formats achieves the best trade-off between MMLU and HellaSwag, and improves performance on 6 out of the 7 transfer tasks. Despite the

target task accuracy, the topic mixture produces the best overall accuracy with 2.1% absolute gain over the random sampling baseline, with excellent transfer to ARC-easy/challenge and OpenBookQA. We note that reweighting $k$-means clusters performs well with an overall 1.6% point improvement, and Table 10 shows that they are the most well-suited for targeting only HellaSwag.

**Topic and format mixtures can be combined** Our domains offer the advantage that topic and format mixtures can be combined. Our experiment (row 5) demonstrates that this is effective, improving performance in 8 out of 9 tasks and achieving a 3.0% absolute gain overall, which narrowly beats the FineWeb-Edu quality classifier. It also consistently improves performance when only aiming for one of the two downstream tasks in Table 10, notably attaining an MMLU score of 33.2%. This illustrates that both topic and format are important axes for data curation.

**Domain mixtures improve quality filters** Finally, we show that our domain mixtures can also boost the overall performance of two state-of-the-art quality filters, improving the average performance of FineWeb-Edu and DCLM-fasttext by 2.0% and 1.0%, points respectively (rows 6-9). We highlight that our domain mixing addresses the weaknesses of the quality classifiers—for instance on HellaSwag, FineWeb-Edu underperforms the random sampling baseline by 1.5% points, which our tailored mixture converts to a 5% absolute gain. This reflects the fact our domain mixtures can be subtly calibrated to meet the demands of the downstream tasks, whereas it would be hard to encode exactly the right preference for certain sub-distributions by changing the prompt for the FineWeb-

Edu classifier or the reference corpus for the DCLM-fasttext classifier. With the same domain reweighting, FineWeb-Edu and DCLM-fasttext achieve similar performance across tasks.

## 5. Quality Filters as Implicit Domain Mixers

In Section 4, we combine domain mixing and quality filtering by using the mixture to specify how many tokens to select per subset. Without an explicit domain mixture, a quality filter will naturally upsample certain domains, which is equivalent to applying an *implicit domain mixture* and subsequently selecting the top documents within each domain. This process offers insights on two quality classifiers considered in this work, and presents a richer way to describe differences between them.

We reconstruct the implicit domain mixture by computing the domain statistics of the quality filtered training datasets. Figure 4 visualizes these distributions for FineWeb-Edu and DCLM-fasttext. We observe that FineWeb-Edu deviates more strongly from the corpus than DCLM-fasttext in terms of topics, while DCLM-fasttext amplifies a larger number of categories in terms of formats. Their mixtures also share notable similarities with the RegMix predictions in Figure 3—all amplifying *Politics*, *Health*, *Science & Tech.*, and *History*, to varying degrees, as well as *Knowledge Articles*, *Tutorials*, *Academic Writing*, and *Q&A Forums*. However, the exact proportions differ substantially and their behaviors also diverge. For example, DCLM-fasttext retains by far the most documents from *Entertainment* and *Games* topics, as well as from *Comment Sections* and *Creative Writing* formats.

**Approximating quality filters by domains** We adopt only the implicit domain mixtures of quality classifiers for pre-training data curation, replacing the "local" selection within each domain with random sampling. The results of training 1B parameter models are shown in Table 2. Both topic and format domains help to approximate the performance of quality classifiers. Of the two quality classifiers under study, we find FineWeb-Edu to be better approximated by domain effects. In this case, implicit Topic × Format mixture recovers its performance gains by 73% on MMLU and 84% on average. However, a substantial gap remains for approximating DCLM-fasttext, suggesting that this classifier relies more on selecting the "right" documents within each domain.

Finally, we report the held-out perplexity of the models and observe that the values for domain mixing and substantially lower than using the quality filtering and close to the baseline corpus. This suggests that document-level quality filtering is a far stronger intervention on the pre-training distribution than rebalancing domains or topics.

Table 2: Approximating quality filters by their implicit mixtures over topics and formats. Numbers in parentheses show how much of the gain of the quality classifier is achieved by mixing alone.

| Data Curation | PPL | MMLU | Task Avg |
|---|---|---|---|
| Baseline corpus | 12.1 | 30.3 | 51.6 |
| FineWeb-Edu | 14.7 | 34.3 | 54.2 |
| as Topic | 12.6 | 32.5 (55%) | 52.4 (29%) |
| as Format | 12.3 | 32.8 (63%) | 52.5 (33%) |
| as Topic × Format | 12.9 | 33.2 (73%) | 53.8 (84%) |
| DCLM-fasttext | 14.0 | 33.4 | 55.1 |
| as Topic | 12.2 | 31.5 (41%) | 51.8 (9%) |
| as Format | 12.2 | 31.4 (35%) | 52.0 (16%) |
| as Topic × Format | 12.5 | 32.0 (56%) | 52.8 (35%) |

**Nature of data quality** It has become common to claim that datasets which produce better benchmark scores have "higher quality" (Li et al., 2023a; Wettig et al., 2024; Sachdeva et al., 2024; Penedo et al., 2024; Li et al., 2024). In domain mixing, the "quality" of a domain is reflected by how much it should be upsampled, but our findings in Section 3 suggest that MMLU and HellaSwag exhibit very different domain preferences, and optimizing for both tasks requires making trade-offs. This highlights how "data quality" is sensitive to the choice of downstream tasks, and we observe that the notion of "quality" by FineWeb-Edu is particularly biased to specific domains that benefit downstream tasks. However, there are many aspects of web content that are not captured by WebOrganizer, e.g., the prevalence of misspellings or factual errors, and these might be better modeled by scoring individual documents. Such effects may explain why DCLM-fasttext is not well approximated by domain effects, and why both quality filters substantially outperform random sampling when imposing the same Topic × Format mixture (Table 2).

## 6. Related Work

**Data selection** Many methods have been developed selecting pre-training data for training large language models. It has become common practice to remove noisy web sites using heuristic filtering rules (Raffel et al., 2020; Rae et al., 2021; Penedo et al., 2023), focusing on surface statistics such as mean word length or word repetitions. This is typically followed by deduplication (Lee et al., 2022; Jiang et al., 2023; Abbas et al., 2023; Tirumala et al., 2023; Soldaini et al., 2024). Additional data selection techniques include measuring n-gram similarity to high-quality reference corpora (Brown et al., 2020; Xie et al., 2023b; Li et al.,

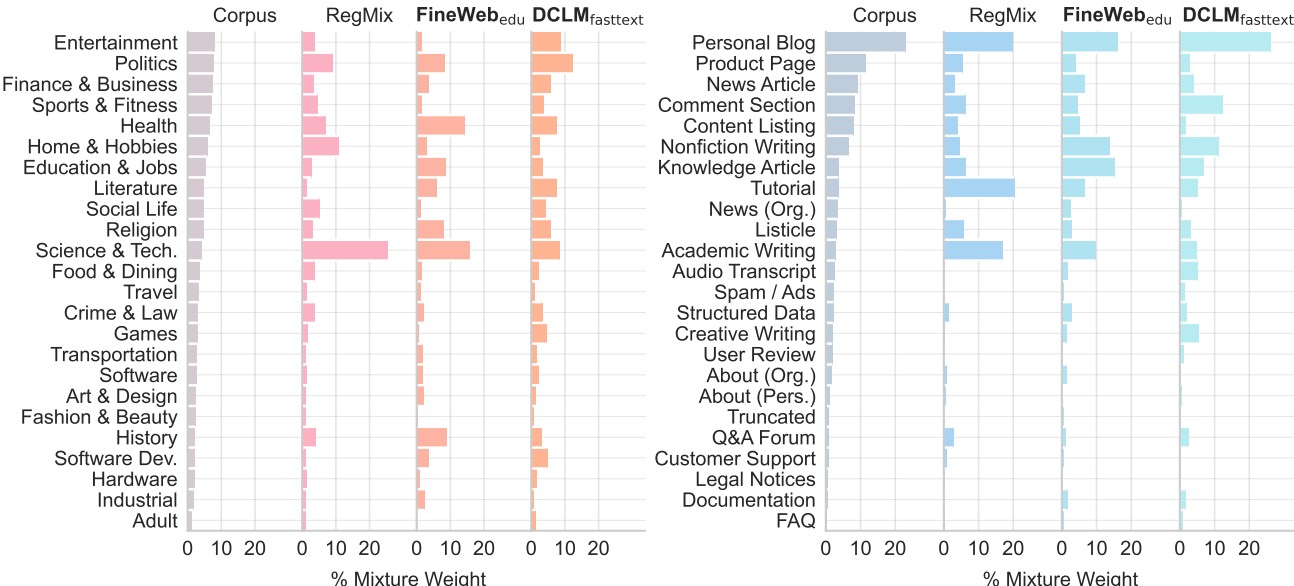

Figure 4: The implicit domain compositions from quality filtering compared to the corpus distribution for topic domains (left) and format domains (right). We include the RegMix prediction tailored to both MMLU and HellaSwag from Figure 3 to facilitate comparison. Numerical values can be found in Table 9 in the appendix.

2024; Brandfonbrener et al., 2024), using perplexity of existing language models (Wenzek et al., 2020; Muennighoff et al., 2023; Marion et al., 2023; Ankner et al., 2025), and prompting large language models to rate documents based on qualities such as factuality or educational value (Gunasekar et al., 2023; Wettig et al., 2024; Sachdeva et al., 2024; Penedo et al., 2024). An alternative approach to data curation focuses on generating synthetic training data from a large language models (Gunasekar et al., 2023; Li et al., 2023a), which may involve designing an explicit taxonomy of skills and concepts (Ding et al., 2023; Ben Allal et al., 2024) or re-writing existing data (Maini et al., 2024)—the latter would reflect the original domain proportions and our approach could be used to improve the data mixture.

**Data curation with domains** Several language models add specially curated domains to their pre-training data (Touvron et al., 2023; Soldaini et al., 2024; OLMo et al., 2024), but CommonCrawl data forms typically the majority of data and has also been shown to outperform domain curation (Penedo et al., 2023; Li et al., 2024). Several works have investigated the specific impact of varying the proportion of code in the pre-training data (Ma et al., 2024; Petty et al., 2024; Aryabumi et al., 2025; Chen et al., 2025). Dubey et al. (2024) briefly mention using knowledge classifiers to downsample "over-represented" data for training Llama-3. Instead of using domains for rebalancing data, Gao et al. (2025) observe performance improvements when conditioning on domain metadata during pre-training. Similar to our work, (Bai et al., 2024) propose to classify data

into 13 topics, and combine this with FineWeb-Edu quality buckets and SlimPajama sources, resulting in a large set of composite domains. Rather than learning the relationship between these domain weights and downstream tasks with RegMix, they propose a selection strategy that samples from the domains based on gradient influence scores. Similarly, (Zhang et al., 2025) define fine-grained $k$-means clusters ($k$=10,000) with the motivation to increase diversity during influence-based data selection. SemDeDup (Abbas et al., 2023) also utilizes a large set of $k$-means clusters to define prototypicality scores and diversify the pre-training distribution. While all these works have the shared goal of improving data quality, our work also contributes a two-dimensional structuree for organizing web data and provides a comparative study of fine-grained quality selection and coarse-grained domain mixing.

**Data mixture optimization** Many techniques have been developed for tuning domains proportions while training language models. These seek to minimize the validation losses across the domains (Xie et al., 2023a; Albalak et al., 2023; Jiang et al., 2024; Chen et al., 2024), although some methods apply to out-of-domain settings (Chen et al., 2023; Fan et al., 2024). Most methods adjust mixtures dynamically during training, and some make predictions from many static mixtures (Liu et al., 2024; Ye et al., 2024; Kang et al., 2024). In concurrent work, Held et al. (2025) use large language model to predict the utility of subsets to downstream tasks. Due to the lack of meaningful domains, CommonCrawl is partitioned into "head" and "tail"

domains based on perplexity scores. Thrush et al. (2025) partition a web corpus into almost 10k domains based on frequent URL domains and rank them based on correlations between domain perplexities and benchmark scores from 90 existing open language models. Hayase et al. (2024) extracts the data mixture of a private pre-training corpus from tokenization rules. Instead of optimizing domain mixtures, researchers have also developed approximations for the impact of individual training examples on loss of a validation set (Engstrom et al., 2024; Yu et al., 2024; Wang et al., 2024). However, (Zhang et al., 2025) demonstrate that partitioning the dataset into domains (in their work, $k$-means clusters) is beneficial for increasing data diversity when selecting data with gradient-based influence approximations.

**Analysis of pre-training data** WebOrganizer can serve as a tool for analyzing the contents of web corpora and the effects of quality filtering. In related work, Longpre et al. (2024b) study data curation in terms of toxicity, source composition, and dataset age, and Longpre et al. (2024a) analyze licensing issues in web corpora. Elazar et al. (2024) provide a scalable tool for searching web-scale corpora and study the prevalence of toxicity, duplicates, and personally identifiable information. Lucy et al. (2024) use self-descriptions of website creators to measure how quality filters amplify and suppress speech across topics, regions, and occupations. Ruis et al. (2024) employ influence functions to find pre-training documents important for learning factual knowledge and mathematical reasoning respectively. In a separate line of work, large language models have been used for clustering large corpora (Wang et al., 2023; Zhang et al., 2023; Pham et al., 2024) and describing clusters post-hoc (Zhong et al., 2022; Tamkin et al., 2024).

## 7. Conclusions

We introduce WebOrganizer—a tool for organizing unstructured web corpora into topic and format domains. By annotating a 200B token pre-training corpus, we demonstrate how WebOrganizer documents the internal contents of the pre-training data, and that we can re-balance these subsets to increase the performance of downstream tasks. Importantly, we show that topic and format selection can be combined, and that domain mixing can be integrated with quality filtering, which combines the benefits of document-level selection with well calibrated domain ratios.

Increasing the transparency of data curation is an interesting avenue for future work. Better documentation of pre-training the data can inform model developers about potential strengths and weaknesses of the model and also improve the understanding of other stakeholders such as policy makers or end users. In this work, we make initial progress in this direction by introducing WebOrganizer and

two high-level domain taxonomies. This enables analyzing the internal composition of web-crawled pre-training corpora (as in Figure 1) and examining how it changes after quality filtering (in Figure 4). There is wide scope for refining these data representations in future work, including hierarchical taxonomies, e.g., breaking down *Science & Technology* into the various scientific disciplines; or multi-label classification which could better account for ambiguous cases where a document covers multiple topics or does not fit cleanly to any one label.

## Impact Statement

Our work advances data curation for language models, and thus carries the broader societal implications associated with improving the capabilities of these models. By taxonomizing web data, we develop a tool that aims to enhance the transparency of pre-training corpora—potentially helping both researchers and the broader public develop a better grasp of the available pre-training data for language models. At the same time, we acknowledge the risks inherent in this process: Reducing the rich diversity of online content to a limited set of discrete domains can obscure important phenomena and may lead to errors, biases, or misrepresentations. We highlight that there are many valid ways to define web taxonomies, and our efforts do not represent a definite "ground truth". Similarly, the predictions of how to rebalance the domains are sensitive to noise, as they are based on relatively few small model runs. Furthermore, is uncertain how well they transfer across model scales. Despite these challenges, in the absence of other meaningful metadata, we believe that our domain annotations contribute to a more informed understanding of web-scale training data.

## Acknowledgments

We thank Pang Wei Koh, Tyler Murray, and Mayee Chen for helpful discussion. We also thank Mengzhou Xia, Dan Friedman, Tianyu Gao, Alex Fang, Maria Antoniak, Ben Lee, and Catherine Chen for feedback on the draft. This research is partially funded by the National Science Foundation (IIS-2211779).

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

# A. Domain Descriptions

Table 3: Detailed overview of our **topic definitions**. We mention common sub-topics for specific categories (e.g., Architecture under Art) and discuss ambiguous cases to decrease the uncertainty when prompting a model and arrive at sharper domain boundaries.

| Topic | Notes |
|---|---|
| Adult | |
| Art & Design | - Includes: architecture |
| Crime & Law | - Includes: law enforcement
- Financial crime and litigation fall under 'Finance & Business'
- Social issues and the legislative process fall under 'Politics' |
| Education & Jobs | - Includes: pedagogy, training & certification, academia
- Educational pages about a topic, e.g., food or mathematics, fall under that topic |
| Entertainment | - Includes: music, movies, TV shows, videos, celebrities, humor, nightlife
- Music or film discussed as art rather than entertainment falls under 'Art & Design' |
| Fashion & Beauty | - Includes: clothing, accessories, cosmetics |
| Finance & Business | - Includes: taxes, regulations, investments, insurance, credit cards, personal finance, corporate communication, marketing, human resources |
| Food & Dining | - Includes: recipes, groceries, beverages, restaurants
- Nutritional sciences fall under 'Health' |
| Games | - Includes: video games, board games, gambling |
| Hardware | - Includes: computer hardware, phones, televisions, other consumer electronics |
| Health | - Includes: medicine, wellness, mental health, veterinary science, nutritional science
- Health insurance falls under 'Finance & Business' |
| History | - Includes: geography, archaeology |
| Home & Hobbies | - Includes: real estate, renting, relocation, furniture, appliances, home improvement, DIY, gardening, pets, toys, collecting |
| Industrial | - Topics related to mining, agriculture, manufacturing, utilities and construction
- Includes: raw materials, industrial goods, chemicals, textiles
- General business topics or business finance fall under 'Finance & Business' |
| Literature | - Includes: literary criticism, linguistics, philosophy, related subjects in the humanities
- Text written in literary style fall under the topic of its contents |
| Politics | - Includes: social issues, political campaigns, the legislative process, geopolitics, protests, activism |
| Religion | - Includes: spirituality |
| Science & Technology | - Includes: physics, chemistry, biology, environmental science, mathematics, statistics, biotech, engineering |
| Social Life | - Includes: family, friends, relationships, community
- Specific social activity (e.g., sports or board games) fall under those topics |
| Software | - Topics related to the use of software and the internet |
| Software Development | - Includes: algorithms, coding, and web development |
| Sports & Fitness | - Includes: martial arts, motor sports, outdoor activities, sports equipment |
| Transportation | - Includes: cars and other vehicles, taxis, public transportation, traffic, commuting, aviation, rail, shipping, logistics |
| Travel | - Includes: hospitality, hotels, sight-seeing, cruises
- Detailed descriptions of tourist destinations fall under 'History' |

Table 4: Detailed overview of our **format definitions**. We mention typical features of these formats to help a model without HTML access deduce the format from the text.

| Format | Notes |
|---|---|
| About (Org.) | - An organizational "About Page", typically containing a self-description or introduction by an organization such as a company, university, government agency, non-profit
- Note that the content may appear similar to a 'Knowledge Article' in some cases, but is not verified and may contain self-promotion |
| About (Personal) | - An "About Page" on a personal website or hobby website, typically containing a self-description, introduction or profile information |
| Academic Writing | - Examples: a research paper, a paper abstract, a thesis, a literature review |
| Audio Transcript | - A written record of spoken language
- Examples: interviews (e.g., in a newspaper), the transcript of a court hearing, movie, podcast, lecture, or speech |
| Comment Section | - A comment section or discussion forum with multiple posts or comments
- Examples: Community sites like reddit, comment sections on news article or blogs |
| Content Listing | - The page contains an overview of content and is used for navigation
- Examples: sitemap, product catalog, search results, news listings with short snippets of articles
- Note that hyperlinks are not visible from the text content and have to be deduced |
| Creative Writing | - The page consists of a short story, chapters from a novel, poem or song lyrics |
| Documentation | - Examples: technical writing, API documentation, README files, source code
- Unlike 'Customer Support', meant for developers and experts, rather than end-users |
| FAQ | - The page content is in the Frequently Asked Questions format |
| Knowledge Article | - Written in an objective and neutral style
- Published on a moderated platform (like Wikipedia) or by a reputable source |
| Legal Notices | - Examples: terms of service, legal disclaimers, privacy policy, license agreement |
| Listicle | - A blog or article that presents content in the form of a list
- Examples: Buzzfeed-style articles, "Top 10" lists, "4 best places to visit in X"
- Lists showing the site contents and facilitate navigation fall under 'Content Listing' |
| News (Org.) | - Organizational news and announcements
- Examples: a press release, a blog post by an organization such as a company, university, government agency, non-profit organization |
| News Article | - Written by journalists on current events and published by news organizations
- Long reads, profiles, editorials, and journalistic essays fall under 'Nonfiction Writing'
- Newspaper interviews fall under 'Audio Transcript' |
| Nonfiction Writing | - Long reads, profiles, editorials, essays, obituaries, memoirs and other forms of nonfiction writing, written by journalists and other professional writers |
| Personal Blog | - Written by an individual typically relating personal experiences and opinions |
| Product Page | - Typically contains descriptions and promotions for a product or service
- Also includes products in a wider sense, for example university course descriptions |
| Q&A Forum | - A user forum with an explicit question & answer format, e.g., Quora, Stack Exchange |
| Spam / Ads | - The page consists primarily of spam content, SEO keyword stuffing, or short online ads for other pages, products or services, or has no apparent purpose |
| Structured Data | - Multiple data entries with a common structure
- Examples: a table, datasheet, movie database, glossary, dictionary, json file, csv, xml |
| Customer Support | - Content by an organization and for a general audience
- Examples: a troubleshooting guide |
| Truncated | - The page contents are incomplete, e.g., truncated, pay-walled, or require a login
- If the page has multiple snippets of truncated articles, choose 'Content Listing'
- Also includes multimedia web pages where the web page text primarily describes and supplements the audiovisual content, e.g., a video description or image gallery |
| Tutorial | - Examples: cooking recipes, DIY instructions, WikiHow page, Khan Academy course
- The page must contain the actual content of the tutorial / how-to guide
- Guides specific to products/services from the website fall under 'Customer Support' |
| User Review | - Reviews posted by users, e.g., on Yelp, TripAdvisor |

**Full prompt**    We provide descriptions of the domains in Table 3 and Table 4. These domain descriptions are given to the model as part of the prompt (with minor adjustments in phrasing). The prompt template is shown in Table 5 and contains instructions, the text contents and URL of the web page, and the the list of domain descriptions. We randomly permute the order in which we list the domains for every new document, and enumerate the randomly shuffled choices as:

```
A: {domain description 1}
B: {domain description 2}
...
```

The random order avoids spurious positional bias from the large language model, and the alphabetic IDs are useful for obtaining single-token outputs from the model, and we use normalize the next-token probabilities of the characters A-X to obtain a soft prediction of domain categories that reflects model uncertainty. We truncate the text contents of web pages at 50K characters and add a truncation hint to the model. We also provide 5 few-shot examples to the model, formatted as previous conversation turns with the same prompt format. Each example is carefully curated to be an interesting case of potential domain conflict and provides an explanation of how the conflict should be resolved. The few-shot examples are also presented in a random order for each annotation.

Table 5: The prompt template for classifying the topic and format of a web page. The first two row shows the templates for system and user prompts, in which {domain} becomes either "topic" and "format" and {instructions} are substituted with the content of the bottom two rows.

| Prompt templates | |
| --- | --- |
| System | `Your task is to classify the {domain} of web pages into one of the following 24 categories:`
`{choices}`

`{instructions}` |
| User | `Consider the following web page:`

`URL: '{url}'`
`Content:  '''`
`{text}`
`'''`

`Your task is to classify the {domain} of web pages into one of the following 24 categories:`
`{choices}`

`{instructions}` |
| **Instructions** | |
| Topic | `Choose which topic from the above list is the best match for describing what the web page content is about.  If the content is about multiple topics, choose the one that is most prominent.  Remember to focus on the topic, and not the format, e.g., a book excerpt about a first date is related to 'Social Life' and not 'Literature'.  The URL might help you understand the content.  Avoid shortcuts such as word overlap between the page and the topic descriptions or simple patterns in the URL. Start your response with the single-letter ID of the correct topic followed by an explanation.` |
| Format | `Choose which format from the above list is the best match for describing the style, purpose and origin of the web page content.  If the content has multiple formats, choose the one that is most prominent.  Remember to focus on the format, and not the topic, e.g., a research paper about legal issues does not count as 'Legal Notices'.  The URL might help you understand the content.  Avoid shortcuts such as word overlap between the page and the format descriptions or simple patterns in the URL, for example '.../blog/...'  may also occur for organizational announcements, comment sections, and other formats.  Start your response with the single-letter ID of the correct format followed by an explanation.` |

# B. Training Domain Classifiers

**Data annotation**   We obtain training data by prompting Llama models to annotate web pages using the prompts described in Appendix A. This includes randomizing the order in which domain descriptions and few-shot examples are presented to the model for each annotation. For all annotations, we leverage the SGLang inference framework (Zheng et al., 2024), and obtain soft probabilities over all category labels by normalizing the next-token probabilities over the alphabetical category labels. We sample web pages for annotations from the RefinedWeb reproduction released by DataComps-LM (Li et al., 2024)—which undergoes similar pre-processing steps as our 200B token pre-training corpus (RefinedWeb filtering and deduplication). For the first stage of training, we annotate 1M web pages with Llama-3.1-8B-Instruct, and for the second stage, a subset of 100K web pages is annotated with Llama-3.1-405B-Instruct, using FP8 inference and 8x H100 NVIDIA GPUs. In both datasets, we reserve the same set of 20K web pages as validation and test sets, therefore leaving 80K annotations for the second phase of training. We repeat the annotation process for both the topic and format taxonomies, and train two separate domain classifiers.

**Fine-tuning setting**   We fine-tune a gte-base-en-v1.5 embedding model, a 140M parameter embedding model, which reports strong performance on benchmarks for a small model and also features a 8192 token context window (Li et al., 2023b), allowing us to process longer documents. In each training stage, we train for a total of 5 epochs with a total batch size of 512 sequences, a learning rate of 1e-4 which is warmed up for the first 10% of training steps and linearly decayed. Our main domain classifiers are shown the same web page features as the prompted Llama models, i.e., the text contents and web page URL, using the template of ``{url}\n\n{text}''. However, for the potential use case of annotating other documents without URL information, we also produce a version of the domain classifiers trained with only the website test as input.

**Classifier accuracy**   We consider how well the domain classifiers imitate the annotations by the Llama-3.1-405B-Instruct models on the validation set of 10K web pages and focus on the subset where the large language model chooses a category with at least 75% confidence—which is the case for 86% of topic annotations and 79% of format annotations. On this subset, we report both the overall accuracy and the worst-group accuracy, i.e., the worst accuracy when predicting a certain label. The results are shown in Table 7. We make the following observations: (1) 2-stage training is particularly effective for improving the worst-group accuracy of the classifiers, and (2) it slightly helps to provide the web page URL to the domain classifiers. Despite these efforts, we note that there remains a gap between the 150M parameter domain classifiers and the 405B parameter Llama-3.1-Instruct model. However, we note that the ceiling for the domain classifier is not 100%. Llama-3.1-Instruct-405B is sensitive to the order in which categories and few-shot examples are presented, and a different random seed produces an agreement of only 98% and 97% on this validation subset for topic and formats respectively, suggesting that the domain classifier introduces an additional 4.4%-5.1% error into the annotation process.

**Domain analysis**   Figure 6 shows the full matrix of normalized PMI scores between topic and format annotations, computed across the 200B token annotated pre-training corpus. Figure 7 visualizes the normalized PMI values between $k$-means clusters and either the topic or the format domains. In Figure 5, we visualize the frequency of URL domains to highlight the need for meaningful coarse-grained domains.

# C. RegMix Implementation

**Sampling training mixtures**   For each domain definition, we generate 512 random domain mixtures for training small models. The mixtures are sampled in a similar fashion to the official RegMix implementation (Liu et al., 2024). We compute the domain proportions in the pre-training corpus and soften the distribution by applying a temperature of $\tau = 2$ to obtain the prior distribution $\boldsymbol{p}$. We then sample training mixtures $\boldsymbol{\pi}$ hierarchically via $\log \alpha \sim \mathrm{Uniform}(\log 0.1, \log 10)$ and $\boldsymbol{\pi} \sim \mathrm{Dirichlet}(\alpha \boldsymbol{p})$.

Table 6: Hyperparameters for small model training

| Parameter | Value |
|---|---|
| Hidden size | 512 |
| Intermediate size | 1536 |
| Activation function | SwiGLU |
| Attention heads | 8 |
| Num. blocks | 8 |
| RoPE base frequency | 10000 |
| Peak learning rate | 3e-3 |
| Cosine cooldown | 3e-4 |
| Warmup ratio | 10% |
| Adam $\beta$'s | (0.9, 0.95) |
| Batch size | 128 |

**Small model training**   We sample 1B tokens according to each training mixture and train small 50M parameter models on this data. The data is tokenized with the GPT-NeoX tokenizer (Black et al., 2022), as used by the DCLM model runs. The model architecture is based on the Llama architecture Touvron et al. (2023), featuring SwiGLU activations (Shazeer, 2020) and RoPE positional embeddings (Su et al., 2024). The 512 model runs require approximately 360 NVIDIA H100 hours. The hyperparameters are given in Table 6.

**Simulation**   We follow Liu et al. (2024) and train a boosted tree regression model to predict downstream loss from the training mixture weights. However, our implementation diverges in the so-called "simulation phase" which seeks to predict the best performing mixture. Liu et al. (2024) generate $N = 1M$ random mixtures according to $\boldsymbol{\pi} \sim \mathrm{Dirichlet}(\boldsymbol{p})$, where $\boldsymbol{p}$ is the prior domain distribution in the corpus (without applying temperature here). The regression model is used to predict a loss for each mixture, and Liu et al. (2024) average $K = 100$ mixtures with the lowest loss to produce a prediction for the best mixture. In our exploration, we encountered the issue that the predicted mixture would be sensitive to the random seed and the hyperparameters $N$ and $K$, and it was also not clear how $K$ should vary when increasing $N$. Liu et al. (2024) do not provide a clear motivation for averaging, but it likely reflects a prior towards smoother distributions. We found that it was more convenient to express this by adding a soft KL constraint to the objective, encouraging the prediction to remain closer to the corpus distribution, $\gamma \mathrm{KL}(\boldsymbol{p} \,||\, \boldsymbol{\pi})$, where the coefficient $\gamma$ is independent of $N$. We also found that increasing $N$ led to diminishing returns and developed a multi-step adaptive search method, which reliably identifies better mixtures under the regression mode. In each iteration, the algorithm updates the prior for generating mixtures with the best current candidate mixture. Algorithm 1 lays out the algorithm. We use the hyperparameters $N = 0.5M$ mixtures, $T = 15$ steps, $\gamma = 0.002$, $\eta = 0.2$ and run the simulation with two random seeds, choosing the better mixture according to the objective. We make a final modification to RegMix when targeting two downstream tasks, i.e., HellaSwag and MMLU. In this case, we fit two separate regression models for the two tasks, and combine them by averaging their outputs.

---

**Algorithm 1** Adaptive search for RegMix

---

**Input:** corpus prior $\boldsymbol{p}$, num. mixtures $N$, KL coefficient $\gamma$, steps $T$, smoothing $\eta$, regression model $f$
**Output:** predicted mixture $\tilde{\boldsymbol{q}}$
1:  $\tilde{\boldsymbol{q}} \leftarrow \boldsymbol{p}$                                                                             ▷ Best mixture overall
2:  $\boldsymbol{w} \leftarrow \boldsymbol{p}$                                                             ▷ Soft average of best mixtures in each iteration
3:  **for** t in 1..T **do**
4:      $\log \alpha^{(i)} \sim \mathrm{Uniform}(\log 1, \log 1000), \quad i \in 1..N$
5:      $\boldsymbol{\pi}^{(i)} \sim \mathrm{Dirichlet}(\alpha^{(i)} \boldsymbol{w})$, s.t. $\boldsymbol{\pi}^{(i)} \leq 6.5\boldsymbol{p}^{(i)}$              ▷ No repetitions when selecting 30B out of 200B tokens.
6:      $\tilde{\boldsymbol{w}} \leftarrow \arg\min_{\boldsymbol{\pi}^{(i)}} f\left(\boldsymbol{\pi}^{(i)}\right) + \gamma KL\left(\boldsymbol{p} \,||\, \boldsymbol{\pi}^{(i)}\right)$

7:      $\boldsymbol{\pi}^{(j)} \leftarrow \beta_j \boldsymbol{w} + (1 - \beta_j)\tilde{\boldsymbol{w}}, \quad \beta_j \in \mathrm{Linspace}(0, 1, 500)$                        ▷ Line search between $\tilde{\boldsymbol{w}}$ and $\boldsymbol{w}$
8:      $\tilde{\boldsymbol{w}} \leftarrow \arg\min_{\boldsymbol{\pi}^{(j)}} f\left(\boldsymbol{\pi}^{(j)}\right) + \gamma KL\left(\boldsymbol{p} \,||\, \boldsymbol{\pi}^{(j)}\right)$

9:      $\boldsymbol{w} \leftarrow \eta \tilde{\boldsymbol{w}} + (1 - \eta)\boldsymbol{w}$                              ▷ Update search prior with the best current mixture
10:     $\tilde{\boldsymbol{q}} \leftarrow \arg\min_{\boldsymbol{\pi} \in \{\tilde{\boldsymbol{w}}, \tilde{\boldsymbol{q}}\}} f\left(\boldsymbol{\pi}\right) + \gamma KL\left(\boldsymbol{p} \,||\, \boldsymbol{\pi}\right)$                              ▷ Keep track of best mixture so far
11: **end for**

---

**Analysis**   For our mixture predictions, we use all 512 mixtures to train the regression model. In an ablation, we reserve 50 mixtures for evaluations and compute the Spearman correlation between the RegMix predictions and small model

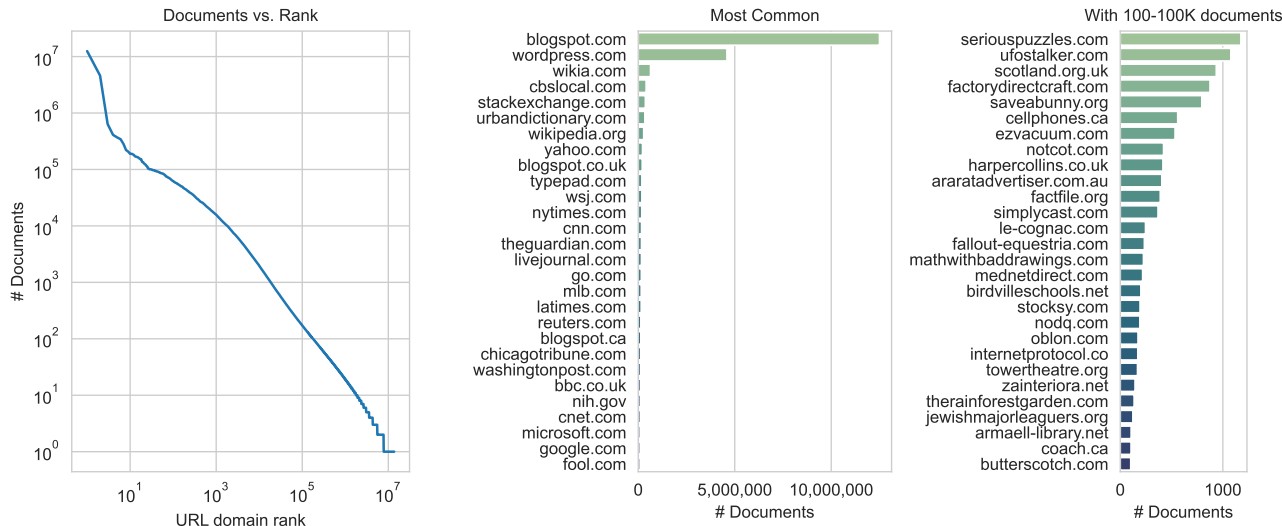

Figure 5: Frequency statistics of URL domain names in our 200B CommonCrawl corpus. Left: Plotting log document frequency vs. the log rank of the domain name exhibits Zipfian long-tail behavior. Middle and right: We list the most common domain names (left) and a random sample of domains between 100-100K documents (right). We plot statistics after removing any sub-domains, i.e. `en.wikipedia.org` → `wikipedia.org`.

evaluations. Table 8 shows the results. The correlation coefficient hovers around 0.90, despite the small size of the models and the out-of-distribution setting of few-shot downstream evaluation. We also explored predicting the held-out distributions using Data Mixing Laws (Ye et al., 2024). However, this achieves worse Spearman correlations and seems overall less stable. We also note that predicting the average loss across MMLU and HellaSwag is slightly more accurate predictions when fitting two separate regression models.

Table 7: The accuracies of domain classifiers to predict confident large language annotations (confidence > 75%). We report both average accuracy and worst-group accuracy.

|  | Topics | | Formats | |
| --- | --- | --- | --- | --- |
|  | Avg | Worst | Avg | Worst |
| Domain classifiers | **93.5** | **87.1** | **91.8** | **80.5** |
| w/o 2-stage training | 91.8 | 84.3 | 90.2 | 74.1 |
| w/o URL features | 92.1 | 86.0 | 88.9 | 80.2 |

# D. Predicted Mixtures

Table 9 reports the numerical results of the mixtures visualized in Figure 3. In addition to our main two tasks of focus, we also include predicted data mixtures for a wider range of downstream tasks in Figure 8. Note that we use bits-per-bytes of the correct solution across all tasks. We use 5 in-context examples for MMLU, HellaSwag, Natural Questions (NQ), 3 examples for HumanEval and MBPP, 0-shot for MATH. We note that both coding tasks upsample documents from the *Software Engineering* topic and *Documentation* format. Natural Questions also exhibits distinct patterns as being the only task to upsample the *Entertainment* topic category heavily.

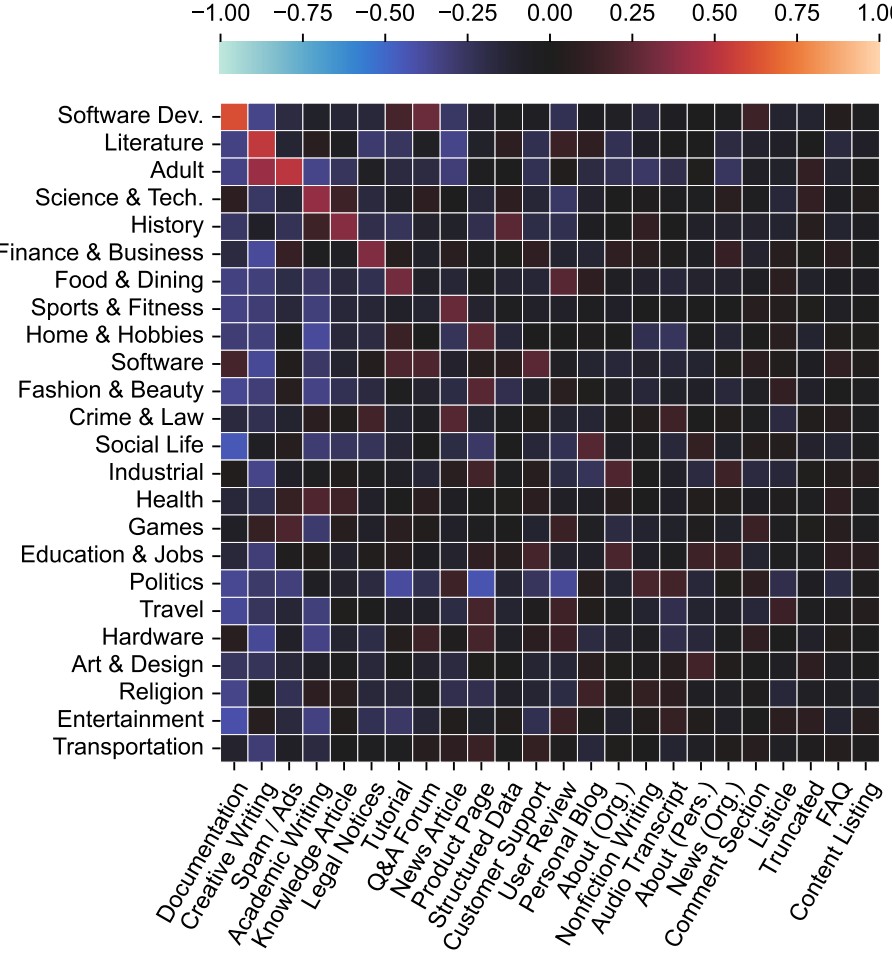

Figure 6: The normalized pointwise-mutual information matrix between all topics (y-axis) and formats (x-axis). A score of 0 indicates independence and 1 implies full co-occurrence.

Table 8: Spearman correlations coefficients when predicting the performance of 50 held-out mixtures using either the LGBM regression model (Ke et al., 2017) or by fitting parametric Data Mixing Laws (Ye et al., 2024). When predicting the average performance of both tasks, we also ablate our approach of fitting separate regression models for each task with the default RegMix setting of having a single regression model predict their average (Liu et al., 2024). Clusters correspond to using $k$-means clusters as domains.

| | Domains | | |
| Target Task | Topics | Formats | Clusters |
|---|---|---|---|
| LGBM Regression | | | |
| MMLU | 0.89 | 0.86 | 0.87 |
| HellaSwag | 0.94 | 0.94 | 0.92 |
| Both | 0.91 | 0.91 | 0.91 |
| *w/ single model* | 0.89 | 0.89 | 0.88 |
| Data Mixing Laws | | | |
| MMLU | 0.79 | 0.84 | 0.70 |
| HellaSwag | 0.80 | 0.91 | 0.82 |
| Both | 0.73 | 0.91 | 0.82 |
| *w/ single model* | 0.64 | 0.89 | 0.83 |

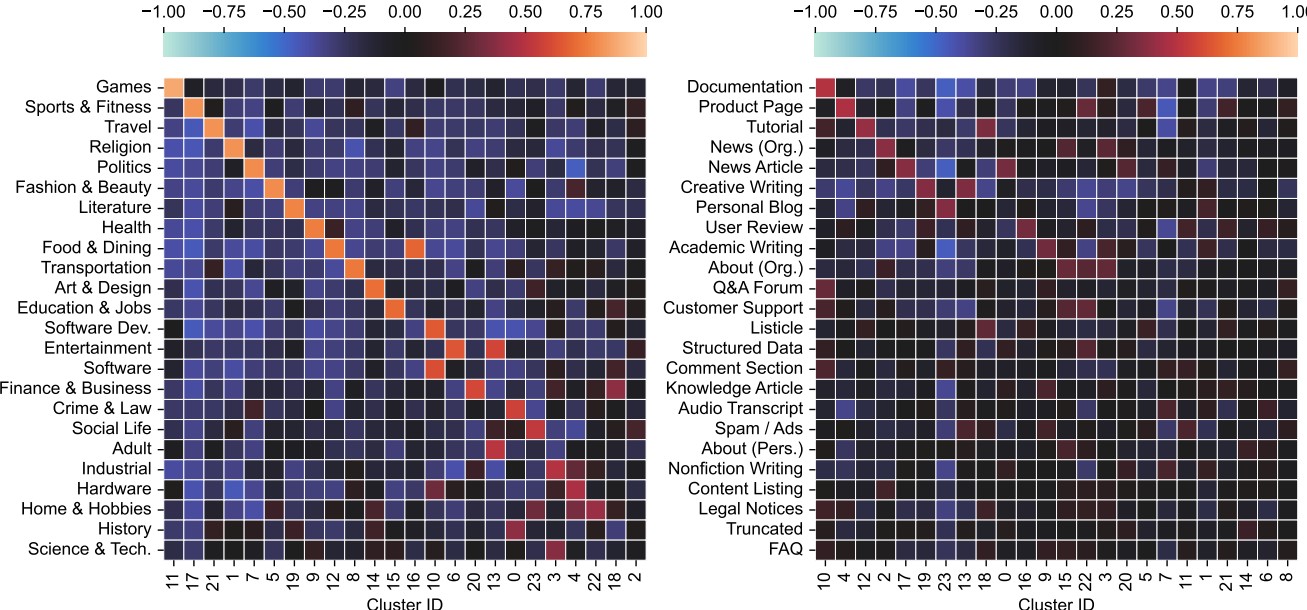

Figure 7: The normalized pointwise-mutual information (NPMI) matrices between $k$-means cluster assignments and the **topic annotations** (left) or **format annotations** (right). A score of 0 indicates independence and 1 implies full co-occurrence. We observe that $k$-means clustering based on document embeddings primarily aligns with topic information.

Table 9: The domain proportions of the corpus, the mixture weights predicted by RegMix, and implicit mixtures used by quality filters. The numbers in parentheses indicate how much the domain is amplified ($>1.0$) or suppressed ($<1.0$) relative to the corpus.

| | Corpus | RegMix | | | Implicit Mixtures | |
|---|---|---|---|---|---|---|
| | | MMLU | HellaSwag | Both | FineWeb-Edu | DCLM-fasttext |
| **Topics** | | | | | | |
| Entertainment | 8.1 | $6.2_{(0.8)}$ | $5.6_{(0.7)}$ | $3.7_{(0.5)}$ | $1.6_{(0.2)}$ | $8.8_{(1.1)}$ |
| Politics | 7.9 | $9.6_{(1.2)}$ | $5.5_{(0.7)}$ | $9.1_{(1.2)}$ | $8.5_{(1.1)}$ | $12.4_{(1.6)}$ |
| Finance & Business | 7.5 | $3.3_{(0.4)}$ | $6.8_{(0.9)}$ | $3.5_{(0.5)}$ | $3.7_{(0.5)}$ | $5.9_{(0.8)}$ |
| Sports & Fitness | 7.3 | $0.1_{(0.0)}$ | $12.3_{(1.7)}$ | $4.5_{(0.6)}$ | $1.5_{(0.2)}$ | $3.8_{(0.5)}$ |
| Health | 6.5 | $14.0_{(2.2)}$ | $5.1_{(0.8)}$ | $7.1_{(1.1)}$ | $14.3_{(2.2)}$ | $7.6_{(1.2)}$ |
| Home & Hobbies | 5.9 | $1.1_{(0.2)}$ | $16.5_{(2.8)}$ | $10.9_{(1.9)}$ | $3.0_{(0.5)}$ | $2.4_{(0.4)}$ |
| Education & Jobs | 5.5 | $4.4_{(0.8)}$ | $3.7_{(0.7)}$ | $2.7_{(0.5)}$ | $8.7_{(1.6)}$ | $3.5_{(0.6)}$ |
| Literature | 4.9 | $1.5_{(0.3)}$ | $1.7_{(0.3)}$ | $1.3_{(0.3)}$ | $5.9_{(1.2)}$ | $7.7_{(1.6)}$ |
| Social Life | 4.8 | $0.1_{(0.0)}$ | $6.1_{(1.3)}$ | $5.1_{(1.1)}$ | $1.3_{(0.3)}$ | $4.4_{(0.9)}$ |
| Religion | 4.8 | $3.7_{(0.8)}$ | $4.9_{(1.0)}$ | $3.2_{(0.7)}$ | $8.1_{(1.7)}$ | $5.7_{(1.2)}$ |
| Science & Tech. | 4.1 | $26.3_{(6.4)}$ | $2.3_{(0.5)}$ | $25.5_{(6.1)}$ | $15.8_{(3.8)}$ | $8.4_{(2.0)}$ |
| Food & Dining | 3.6 | $1.6_{(0.4)}$ | $3.1_{(0.9)}$ | $3.6_{(1.0)}$ | $1.4_{(0.4)}$ | $2.2_{(0.6)}$ |
| Travel | 3.3 | $3.4_{(1.0)}$ | $1.6_{(0.5)}$ | $1.4_{(0.4)}$ | $1.3_{(0.4)}$ | $1.2_{(0.4)}$ |
| Crime & Law | 3.1 | $5.7_{(1.8)}$ | $2.1_{(0.7)}$ | $3.8_{(1.2)}$ | $2.2_{(0.7)}$ | $3.3_{(1.1)}$ |
| Games | 3.0 | $0.9_{(0.3)}$ | $1.8_{(0.6)}$ | $1.6_{(0.5)}$ | $0.6_{(0.2)}$ | $4.7_{(1.5)}$ |
| Transportation | 2.7 | $1.2_{(0.4)}$ | $2.0_{(0.7)}$ | $1.1_{(0.4)}$ | $1.7_{(0.6)}$ | $1.8_{(0.6)}$ |
| Software | 2.7 | $0.1_{(0.0)}$ | $3.2_{(1.2)}$ | $1.3_{(0.5)}$ | $2.0_{(0.7)}$ | $2.3_{(0.8)}$ |
| Art & Design | 2.4 | $2.0_{(0.8)}$ | $1.1_{(0.4)}$ | $1.0_{(0.4)}$ | $2.3_{(0.9)}$ | $1.4_{(0.6)}$ |
| Fashion & Beauty | 2.4 | $0.0_{(0.0)}$ | $4.8_{(2.0)}$ | $1.1_{(0.5)}$ | $0.3_{(0.1)}$ | $0.7_{(0.3)}$ |
| History | 2.3 | $6.7_{(3.0)}$ | $1.4_{(0.6)}$ | $4.1_{(1.8)}$ | $9.0_{(4.0)}$ | $3.2_{(1.4)}$ |
| Software Dev. | 2.2 | $4.1_{(1.9)}$ | $2.2_{(1.0)}$ | $1.1_{(0.5)}$ | $3.6_{(1.6)}$ | $4.8_{(2.2)}$ |
| Hardware | 2.1 | $3.2_{(1.5)}$ | $2.0_{(0.9)}$ | $1.4_{(0.7)}$ | $1.0_{(0.5)}$ | $1.7_{(0.8)}$ |
| Industrial | 1.7 | $0.8_{(0.5)}$ | $1.4_{(0.8)}$ | $0.9_{(0.5)}$ | $2.4_{(1.4)}$ | $0.8_{(0.5)}$ |
| Adult | 1.1 | $0.0_{(0.0)}$ | $2.7_{(2.5)}$ | $0.9_{(0.9)}$ | $0.0_{(0.0)}$ | $1.3_{(1.2)}$ |
| **Formats** | | | | | | |
| Personal Blog | 22.9 | $26.4_{(1.2)}$ | $31.5_{(1.4)}$ | $19.7_{(0.9)}$ | $16.2_{(0.7)}$ | $26.0_{(1.1)}$ |
| Product Page | 11.5 | $6.0_{(0.5)}$ | $7.6_{(0.7)}$ | $5.3_{(0.5)}$ | $4.1_{(0.4)}$ | $2.8_{(0.2)}$ |
| News Article | 9.0 | $4.5_{(0.5)}$ | $7.0_{(0.8)}$ | $3.1_{(0.3)}$ | $6.7_{(0.7)}$ | $3.9_{(0.4)}$ |
| Comment Section | 8.3 | $8.7_{(1.0)}$ | $4.8_{(0.6)}$ | $6.2_{(0.7)}$ | $4.4_{(0.5)}$ | $12.4_{(1.5)}$ |
| Content Listing | 7.9 | $6.9_{(0.9)}$ | $5.6_{(0.7)}$ | $4.1_{(0.5)}$ | $5.2_{(0.7)}$ | $1.6_{(0.2)}$ |
| Nonfiction Writing | 6.6 | $5.4_{(0.8)}$ | $4.7_{(0.7)}$ | $4.5_{(0.7)}$ | $13.7_{(2.1)}$ | $11.0_{(1.7)}$ |
| Knowledge Article | 3.6 | $6.8_{(1.9)}$ | $2.5_{(0.7)}$ | $6.2_{(1.7)}$ | $15.2_{(4.2)}$ | $6.9_{(1.9)}$ |
| Tutorial | 3.6 | $5.3_{(1.5)}$ | $20.2_{(5.7)}$ | $20.3_{(5.7)}$ | $6.7_{(1.9)}$ | $5.1_{(1.4)}$ |
| News (Org.) | 3.4 | $0.7_{(0.2)}$ | $2.2_{(0.7)}$ | $0.6_{(0.2)}$ | $2.6_{(0.8)}$ | $0.6_{(0.2)}$ |
| Listicle | 3.1 | $2.0_{(0.6)}$ | $2.5_{(0.8)}$ | $5.7_{(1.9)}$ | $2.8_{(0.9)}$ | $3.1_{(1.0)}$ |
| Academic Writing | 2.7 | $16.8_{(6.2)}$ | $1.9_{(0.7)}$ | $16.9_{(6.3)}$ | $9.7_{(3.6)}$ | $4.9_{(1.8)}$ |
| Audio Transcript | 2.5 | $0.2_{(0.1)}$ | $1.2_{(0.5)}$ | $0.1_{(0.0)}$ | $1.7_{(0.7)}$ | $5.1_{(2.0)}$ |
| Spam / Ads | 2.2 | $0.0_{(0.0)}$ | $1.3_{(0.6)}$ | $0.0_{(0.0)}$ | $0.5_{(0.2)}$ | $1.4_{(0.6)}$ |
| Structured Data | 2.1 | $1.2_{(0.6)}$ | $1.5_{(0.7)}$ | $1.4_{(0.7)}$ | $2.8_{(1.3)}$ | $1.8_{(0.9)}$ |
| Creative Writing | 1.9 | $0.4_{(0.2)}$ | $1.0_{(0.5)}$ | $0.3_{(0.2)}$ | $1.3_{(0.7)}$ | $5.4_{(2.9)}$ |
| User Review | 1.9 | $0.0_{(0.0)}$ | $1.3_{(0.7)}$ | $0.0_{(0.0)}$ | $0.2_{(0.1)}$ | $1.2_{(0.7)}$ |
| About (Org.) | 1.7 | $2.5_{(1.5)}$ | $1.3_{(0.8)}$ | $0.9_{(0.5)}$ | $1.4_{(0.9)}$ | $0.3_{(0.2)}$ |
| About (Pers.) | 1.1 | $0.6_{(0.6)}$ | $0.3_{(0.2)}$ | $0.5_{(0.5)}$ | $0.2_{(0.1)}$ | $0.4_{(0.4)}$ |
| Truncated | 0.9 | $0.9_{(1.0)}$ | $0.2_{(0.2)}$ | $0.1_{(0.1)}$ | $0.6_{(0.7)}$ | $0.2_{(0.3)}$ |
| Q&A Forum | 0.8 | $2.4_{(3.0)}$ | $0.5_{(0.6)}$ | $2.8_{(3.5)}$ | $1.2_{(1.5)}$ | $2.6_{(3.2)}$ |
| Customer Support | 0.8 | $0.5_{(0.6)}$ | $0.3_{(0.4)}$ | $0.8_{(1.0)}$ | $0.6_{(0.8)}$ | $0.3_{(0.5)}$ |
| Legal Notices | 0.6 | $0.3_{(0.6)}$ | $0.2_{(0.3)}$ | $0.1_{(0.2)}$ | $0.2_{(0.3)}$ | $0.1_{(0.2)}$ |
| Documentation | 0.6 | $1.0_{(1.7)}$ | $0.5_{(0.9)}$ | $0.1_{(0.1)}$ | $1.6_{(2.7)}$ | $1.6_{(2.8)}$ |
| FAQ | 0.4 | $0.4_{(1.0)}$ | $0.1_{(0.2)}$ | $0.1_{(0.4)}$ | $0.5_{(1.3)}$ | $0.9_{(2.4)}$ |

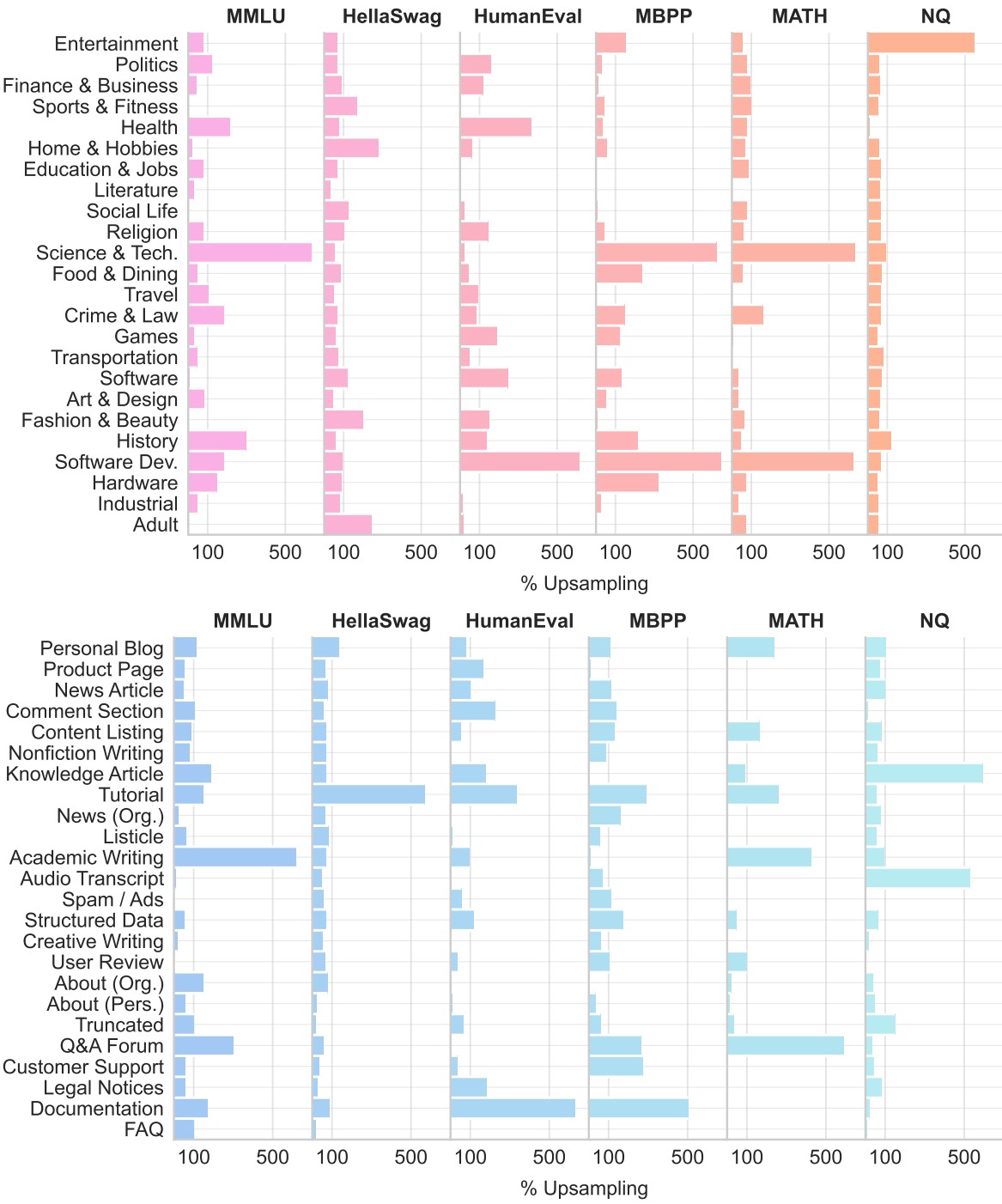

Figure 8: The predicted upsampling by RegMix of our topic domains (top) and formats (bottom), as a proportion of the corpus distributions. Note that the RegMix search constraints the upsampling to a maximum of 650%.

# E. Experimental Details

**Data pre-processing**    We use the `1b-1x` data pool from DataComps-LM (Li et al., 2024) to facilitate comparisons with future work. The raw pool consists of 1.64T tokens, extracted from CommonCrawl with `resiliparse`. We follow the best practice established by (Li et al., 2024) and run heuristic filtering to eliminiate noisy web artifacts, specifically, the set of filters from the RefinedWeb dataset (Penedo et al., 2023). However, to the best of our knowledge, our reproduction differs slightly from Li et al. (2024), since we do not use the "high-quality URL filter", which was originally meant to exclude documents from high-quality domains such as Wikipedia and Github, such that they can be added to the data mix manually. In the next step, we perform deduplication using Bloom filter (Soldaini et al., 2024), while Li et al. (2024) use the MinHash algorithm in their `1b-1x` baseline (Broder, 1997). The resulting corpus contains 200B tokens and constitutes the "token universe" for our data selection experiments and analyses. We annotate this corpus with quality scores from the FineWeb-edu classifier (Penedo et al., 2024) and the DCLM `fasttext OH-2.5 + ELI5` model (Joulin et al., 2017; Li et al., 2024), as well as with the top-1 prediction from the topic and format classifiers, and $k$-means cluster assignments.

**Data selection**    From this 200B token base corpus, we set apart approximately 1B token as a validation set and use the rest for selecting training data. For each training run, we include enough documents to amount to 30B tokens. This is slightly more than the 29B tokens required by DCLM for a `1b-1x` training run, but it ensures that there are enough tokens during model training, since some tokens are dropped in the subsequent tokenization and packing stage. For quality selection, we select the highest scoring documents until the token budget is reached. We speed up the tokenization process by allowing "imbalanced" chunks, decreasing the chunk size to 2048 sequences, and scaling across many workers. The DCLM default choice of balancing chunks before writing was prohibitively slow on the slurm cluster we used.

**Model training**    We use the `1b-1x` reference setting from Li et al. (2024). The models have 1,439,795,200 parameters and are trained for 28,795,904,000 tokens with a batch size of 256 and a sequence length of 2048 tokens. We speed up training by adding `torch.compile`, making a single training run take 183 NVIDIA H100 hours.

**Evaluation setting**    We use the OLMES evaluation framework (Gu et al., 2024) and evaluate on 9 tasks with a 5-shot in-context learning prompt: MMLU (Hendrycks et al., 2021), HellaSwag (HSwag) (Zellers et al., 2019), PIQA (Bisk et al., 2020), WinoGrande (WinoG) (Sakaguchi et al., 2021), CommonSenseQA (CSQA) (Talmor et al., 2019), Social IQa (SIQA) (Sap et al., 2019), ARC-easy/challenge (ARC-e/ARC-c) (Clark et al., 2018), and OpenBookQA (OBQA) (Mihaylov et al., 2018). The OLMES task suite also includes BoolQ (Clark et al., 2019). However, we found that it produced unreliable results, e.g., the random sampling baseline would achieve 63.8%, and DCLM-fasttext selection would 54.4%, which is 9.4 percentage points lower and would have a large impact on the average performance.

We also used the DCLM evaluation framework to measure the `Core` score, a normalized task average across 22 tasks (Li et al., 2024), which we report in Table 10. However, we find that OLMES routinely measures higher accuracies in common tasks (HellaSwag, PIQA, and WinoGrande), which is useful for discriminating between models. We also observed that some `Core` tasks from BigBench and AGI eval are close to random performance at the `1b-1x` scale. Furthermore, given the symbolic nature of some tasks, e.g., dyck sequence completion, MMLU and HellaSwag are likely not good proxies for finding the best domain mixture. Note that we were unable to reproduce the exact *Baseline* and *DCLM-fasttext* performance by Li et al. (2024), likely due to small differences in the data pre-processing stage, as discussed at the start of this section.

Table 10: Detailed results of our data mixing experiments, including the `Core` score from DCLM (Li et al., 2024) and held-out perplexity on the baseline corpus. In each row, we highlight the tasks used to optimize the domain mixture.

| Data Curation | MMLU | HSwag | PIQA | WinoG | CSQA | SIQA | ARC$_e$ | ARC$_c$ | OBQA | Avg | Core | PPL |
|---|---|---|---|---|---|---|---|---|---|---|---|---|
| *Baseline* | 30.3 | 57.5 | 71.3 | 56.1 | 59.0 | 49.9 | 62.2 | 34.0 | 44.0 | 51.6 | 26.1 | 12.1 |
| *Domain mixing: MMLU* | | | | | | | | | | | | |
| Clusters | 32.0 | 57.0 | 70.2 | 55.4 | 59.4 | 50.7 | 64.2 | 36.1 | 43.4 | 52.0 | 26.0 | 12.7 |
| Topic | 32.3 | 52.7 | 68.5 | 56.0 | 57.1 | 48.8 | 70.2 | 38.7 | 44.4 | 52.1 | 26.7 | 12.8 |
| Format | 32.0 | 56.3 | 70.8 | 55.5 | 59.5 | 50.6 | 66.2 | 36.9 | 42.2 | 52.2 | 26.7 | 12.3 |
| Topic × Format | 33.2 | 54.1 | 69.9 | 55.6 | 58.6 | 48.3 | 71.4 | 40.9 | 45.0 | 53.0 | 26.4 | 13.0 |
| *Domain mixing: HellaSwag* | | | | | | | | | | | | |
| Clusters | 30.5 | 61.0 | 74.1 | 57.1 | 61.0 | 49.7 | 64.4 | 34.6 | 42.2 | 52.7 | 25.4 | 12.3 |
| Topic | 30.1 | 60.1 | 72.8 | 56.7 | 57.8 | 47.6 | 63.3 | 32.8 | 39.4 | 51.2 | 24.0 | 12.3 |
| Format | 31.1 | 60.6 | 73.0 | 57.4 | 60.8 | 48.7 | 64.3 | 35.8 | 42.4 | 52.7 | 27.7 | 12.2 |
| Topic × Format | 30.2 | 61.4 | 74.0 | 58.7 | 61.9 | 50.3 | 64.4 | 35.2 | 49.2 | 53.9 | 27.2 | 12.4 |
| *Domain mixing: MMLU and HellaSwag* | | | | | | | | | | | | |
| Clusters | 31.8 | 59.4 | 73.4 | 58.2 | 58.7 | 50.7 | 66.1 | 35.2 | 44.8 | 53.2 | 26.9 | 12.7 |
| Topic | 31.4 | 56.2 | 72.1 | 54.8 | 61.3 | 47.8 | 70.3 | 40.6 | 49.0 | 53.7 | 28.5 | 12.8 |
| Format | 31.7 | 60.9 | 74.1 | 56.9 | 60.1 | 47.4 | 65.8 | 35.9 | 47.6 | 53.4 | 27.1 | 12.5 |
| Topic × Format | 32.7 | 60.1 | 73.4 | 56.5 | 62.3 | 49.3 | 69.7 | 38.8 | 49.0 | 54.6 | 28.2 | 12.6 |
| *Quality filtering (+ domain mixing: MMLU and HellaSwag)* | | | | | | | | | | | | |
| FineWeb-Edu | 34.3 | 56.0 | 69.9 | 57.7 | 60.0 | 47.9 | 71.9 | 42.3 | 48.2 | 54.2 | 29.1 | 14.7 |
| + Topic × Format | 34.2 | 62.5 | 73.3 | 57.1 | 63.0 | 49.4 | 72.2 | 43.3 | 50.8 | 56.2 | 29.8 | 13.8 |
| DCLM-fasttext | 33.4 | 59.0 | 70.5 | 58.8 | 63.2 | 50.7 | 71.4 | 39.8 | 48.8 | 55.1 | 29.4 | 14.0 |
| + Topic × Format | 33.8 | 63.1 | 74.3 | 57.6 | 62.7 | 49.8 | 73.4 | 42.2 | 47.8 | 56.1 | 30.2 | 13.7 |

