# OpenReview forum: "Organize the Web: Constructing Domains Enhances Pre-Training Data Curation"
_ICML.cc/2025/Conference — ICML 2025 poster_

### Official Review · Reviewer_suAs · 2025-02-19

**Overall Recommendation:** 2

**Summary:**

This paper presents WebOrganizer, a method to construct, and annotate two sets of explainable and orthogonal domains (24 topics and 24 formats) with LLMs (e.g., Llama-3.1-405B-Instruct). The annotations are then approximated by a smaller (140M) model, making it scale efficiently to large corpora. It then employs the technique from RegMix to find the optimal domain mixture to optimize target performance by treating MMLU and HellaSwag as reference tasks. Experiment results show that the annotated domains can curate the pretraining data better than semantic clustering, and also augment quality-based filters to further improve the data curation outcome. Additionally, topic and format domains are basically orthogonal and complement each other. Analyses provide further insights into how quality-based filters implicitly change the domain mixture.

**Claims And Evidence:**

Claims are supported by clear evidence, but some experimental and evaluation drawbacks below may make the claims about the improvements over baseline methods less convincing:

- The cluster baselines only group the data into 24 clusters, which is far away from the optimal cluster number (~10k) in previous works [1][2]. I don't think the number of clusters needs to be strictly the same as the number of domains in their comparison.
- The improvements on OLMES (Table 1) may not smoothly transfer to DCLM Core (which is **more informative and comprehensive** from my perspective). For example, the topic × format one cannot outperform a single topic or format in DCLM Core.

[1]: Abbas, Amro, et al. "Semdedup: Data-efficient learning at web-scale through semantic deduplication." arXiv preprint arXiv:2303.09540 (2023).

[2]: Zhang, Chi, et al. "Harnessing Diversity for Important Data Selection in Pretraining Large Language Models." arXiv preprint arXiv:2409.16986 (2024).

**Essential References Not Discussed:**

I think [1] is a paper that has similar contributions to this one, which leverages GPT-4o as a topic annotator and trains a BERT-based model as the topic classifier. Its data selection pipeline also considers both domain, topic, and quality dimensions. I have not found the citation to [1] in this paper and I think the contribution of WebOrganizer seems a little bit limited given the similarity to prior works. **It also publicly releases its BERT topic classifier, which could serve as a direct comparison to the topic classifiers in this paper.**

[1] Bai, Tianyi, et al. "Multi-Agent Collaborative Data Selection for Efficient LLM Pretraining." arXiv preprint arXiv:2410.08102 (2024).

**Experimental Designs Or Analyses:**

The experiment and analysis design is overall sound and self-consistent, except for the cluster baseline/evaluation issues I mentioned above, which I feel are important for my judgment.

**Methods And Evaluation Criteria:**

Their methods to construct, and annotate two sets of explainable and orthogonal domains, and optimize their mixtures to improve target performance overall make sense. It would be better to make the domain construction more automated with fewer human efforts.

In terms of the evaluation criteria, I do have some different opinions. The authors mention in the Appendix that *given the symbolic nature of some tasks, e.g., dyck sequence completion, MMLU, and HellaSwag are likely not good proxies for finding the best domain mixture.*, while I think it is a better practice for this research to adopt better proxy data that can generalize better than MMLU and HellaSwag (e.g., some instruction data), rather than limit the main evaluation to the tasks more related to the chosen proxies. In other words, it is more convincing to choose DCLM Core as the main evaluation rather than OLMES since the main setup of this work is based on DCLM.

**Other Comments Or Suggestions:**

I would also suggest the authors try out different (maybe larger) DCLM setups to show the generalization abilities of their method, but the feasibility definitely depends on the available computational resources the authors have because pretraining is always costly.

**Other Strengths And Weaknesses:**

**Strengths**

- This paper provides an explainable technique to analyze the inherent composition of pretraining data and the behavior of quality filters.
- WebOrganizer has the potential to improve over current individual quality-based filters by identifying the optimal domain weights.
- The writing is clear.

**Weaknesses**

- The proposed method to identify and annotate topic domains, technically, does not significantly vary from prior works (please refer to Essential References Not Discussed).

- The choice of the domain taxonomies (topic and format) and the number of used domains (24) are somehow intuitively set (or found) by humans. A more fundamental question would be how to identify the taxonomy and granularity of domain classifications more automatically.

- The experiment setups can be polished as well. (1) the full potential of the clustering baseline has not been thoroughly presented; (2) the improvements over a more holistic evaluation set (DCLM Core in Table 10) are not as significant as in Table 1. DCLM is meant to be a standardized benchmark to evaluate the pretraining curation outcome and it is better to follow their evaluation setup. Specifically, the DCLM Core set consists of more than 20 tasks while the 9 evaluation tasks in OLMES seem to favor too much knowledge and commonsense abilities. Furthermore, the authors use the MMLU training sets as the target tasks, which already have the in-domain data from ARC-e, ARC-c, and OBQA.

**Questions For Authors:**

1. What is the clustering performance if we scale the number of clusters from 24 to 10k-level following the best practices in prior works [1][2]?
2. What if we combine clusters x format in the data selection because *clusters mostly align with topics* in terms of both characteristics and evaluation results? Will it match the performance of the topic × format? Also, do you have some insights on why a single topic or format classifier cannot outperform clusters too much?
3. Can you provide a comparison of your topic classifier with the one released by [3]?
4. What is the detailed cost (preferred specific numbers) to adopt RegMix to find the optimal domain mixtures, i.e., training 512 models of 50M parameters for 1B tokens?
5. In Lines 142-143, could you provide more details on how humans and LLM (Llama-3.1-405B-Instruct) work together to refine the topic and format definitions?
6. How do you extract *75% confidence* from LLM annotation?
7. This is just out of curiosity: will different annotation LLMs (like DeepSeek) have different behaviors in terms of classifying domains?

[1]: Abbas, Amro, et al. "Semdedup: Data-efficient learning at web-scale through semantic deduplication." arXiv preprint arXiv:2303.09540 (2023).

[2]: Zhang, Chi, et al. "Harnessing Diversity for Important Data Selection in Pretraining Large Language Models." arXiv preprint arXiv:2409.16986 (2024).

[3] Bai, Tianyi, et al. "Multi-Agent Collaborative Data Selection for Efficient LLM Pretraining." arXiv preprint arXiv:2410.08102 (2024).

**Relation To Broader Scientific Literature:**

This paper is related to previous domain mixing work in pretraining scenarios but extends the concepts of domains from simple web URLs to more fine-grained and explainable topics/formats identified by LLMs, which can provide a new perspective to pretraining data curation. However, the key technical contributions seem a little bit limited due to the existence of similar previous research [1].

[1] Bai, Tianyi, et al. "Multi-Agent Collaborative Data Selection for Efficient LLM Pretraining." arXiv preprint arXiv:2410.08102 (2024).

**Theoretical Claims:**

This paper does not make theoretical claims and focuses mostly on the empirical part.

---

> ### Author Rebuttal · Authors · 2025-04-01
>
> Thank you for your review and for engaging closely with our work! We are glad you found that our methodology is overall sound and that our claims are supported by clear evidence. However, you raised several points with respect to evaluation and novelty, which we hope to address below.
>
> > Comparing to greater number of clusters in SemDeDup and Quad
>
> The central premise of our paper revolves around the question of how to partition a monolithic dataset into meaningful subsets for data mixing. Therefore, we consider different ways of structuring the data into an equal number of partitions and apply a fixed domain mixture optimization algorithm (for which we use RegMix). We believe this is a fair experimental setting given our paper’s goals. However, the reviewer correctly points out interesting future extensions, such as studying the effect of domain granularity.
>
> The two mentioned papers, SemDeDup [1] and Quad [2], are fantastic related works, but neither of them perform k-means clustering with the goal of learning a data mixture. Instead, SemDeDup [1] performs clustering to compute “prototypicality” scores and select data points that are least similar to the cluster center — therefore performing sample-level filtering — and the DCLM paper finds that it performs worse than the RefinedWeb baseline (Table 4).  Quad [2] employs fine-grained clustering to obtain robust and efficient influence estimates across groups and then employs binary thresholding to select groups of similar data points from a corpus. Their proposed algorithm is designed for fine-grained group selection instead of data mixture optimization and requires careful tuning of the alpha and gamma hyperparameters. We agree with the reviewer that it would be interesting to add Quad to the experiments, but we would not think of it as a baseline for domain mixing, but rather compare it to FineWeb-Edu and DCLM-fasttext and see whether Quad would also benefit from reweighting domains. We hope to add these experiments in the final version of the paper.
>
> > DCLM Core is more informative and comprehensive than OLMES
>
> DCLM Core may contain a more comprehensive range of tasks, but in our experiments, we noticed substantial variance with the DCLM Core tasks at the 1B parameter scale (up to 1 point) and many tasks were close to random guessing performance. In fact, the DCLM paper acknowledges that other evaluation frameworks such as LightEval may be more suitable for evaluating small models. We also note that data mixing is a more subtle intervention on the training data than quality filtering, requiring more stable evaluations to reveal small differences between models. Ultimately, we found OLMES to be more reliable and informative. We also note that DCLM Core does not include MMLU, although MMLU is undoubtedly one of the most popular benchmarks for LLMs. We agree with the reviewer that it would be an interesting avenue for future work to investigate better and more diverse proxy data for domain optimization, but this was ultimately not the focus of our work. Instead, we chose MMLU and HellaSwag as they are widely known in the research community and used in almost all previous data selection works.
>
> > It would be better to make domain construction more automated with less human effort
>
> Since pre-training data curation is so crucial in shaping the behavior of a model that may potentially be used by millions of people, we argue that there is some value in retaining human oversight in the data curation process! Nevertheless, we expect future methods to automate the process of taxonomizing and structuring the web with much finer granularity.
>
> > Relation to Bai et al.
>
> Thank you for bringing this interesting paper to our attention! We agree that there are similarities with our work and we will definitely highlight them in the next version of our paper. Most notably, Bai et al. [3] also contribute a topic classifier of 13 topics derived by summarizing k-means clusters, which we will compare to our 24 topics. From our analysis in Section 2.4, we would expect k-means clusters to reflect topic information, whereas it would be harder to obtain format domains this way. Besides contributing a high-quality set of format domains, we believe our work makes several independent contributions: (1) We show that topic and formats are approximately orthogonal and their distributions can be combined, (2) We propose a new, simple and effective way of combining data mixing with quality-based data selection, (3) We study the implicit domain biases of two popular quality filters.
>
> Questions: Unfortunately, we have reached the rebuttal space limit and cannot respond to the interesting questions here
>
> [1] Abbas et al. "SemDeDup: Data-efficient learning at web-scale through semantic deduplication"
>
> [2] Zhang et al. "Harnessing Diversity for Important Data Selection in Pretraining Large Language Models"
>
> [3] Bai et al. "Multi-Agent Collaborative Data Selection for Efficient LLM Pretraining"

---

### Official Review · Reviewer_oDMK · 2025-03-14

**Overall Recommendation:** 4

**Summary:**

The paper introduces WebOrganizer, a framework to categorize pre-training data for LMs using topics and formats. The labels are annotated by a pretrained LM. Experiments show that (1) reorganizing data according to these domain classifications significantly improves downstream performance, (2) integrating WebOrganizer with existing quality-based data filtering methods further boosts model performance, (3) existing quality filters are implicitly doing filtering based on topics and formats.

**Claims And Evidence:**

Most claims made in this paper are generally clear and supported by convincing experiments. One assumption I don't understand is Sec 4.2 "we make the assumption that we can select topics and formats independently", an analysis on a smaller subset of pretraining data could be interesting.

**Essential References Not Discussed:**

N/A

**Experimental Designs Or Analyses:**

The pretraining and evaluation designs (for both domain mixing and its combination with quality filters) sound valid to me.

**Methods And Evaluation Criteria:**

Most methods and evaluation are adopted from existing works (e.g., pretraining data, domain importance estimation methods, benchmarks). They are pretty standard and make sense for the problem.

**Other Comments Or Suggestions:**

N/A

**Other Strengths And Weaknesses:**

N/A

**Questions For Authors:**

N/A

**Relation To Broader Scientific Literature:**

The paper is clearly related to existing literature on pre-training data selection and quality filtering. It shows that (1) a simple domain-based filtering is more effective than quality-based filtering, (2) domain-based selection can improve quality-based filtering.

**Theoretical Claims:**

The paper does not include significant theoretical claims or formal proofs. I don't fully understand the assumption in Sec 4.2 "we make the assumption that we can select topics and formats independently".

---

> ### Author Rebuttal · Authors · 2025-04-01
>
> Thank you for your kind review! We are glad you found the claims in the paper clear and supported by convincing experiments!
>
> > Assumption of selecting topics and formats independently
>
> We don’t mean to make an assumption in the theoretical sense, but we simply propose to form a mixture distribution of p(topic, format) = p(topic) p(format) given individual mixture distributions p(topic) and p(format). This is motivated by Section 2.3, where we find that topics and formats often have close to zero pointwise mutual information, i.e., they co-occur almost as though they were independently distributed.

---

### Official Review · Reviewer_jae1 · 2025-03-14

**Overall Recommendation:** 3

**Summary:**

This paper tackles two main challenges: first is to separate the huge pre-training corpus into groups of domains and styles, and the second part is to compute the optimum mixing/reweighting strategy to combine these datasets to achieve targeted improvements in few specific metrics like MMLU and HellsSwag. They also demonstrate that this approach is potentially to quality filtering with complementary gains using both. Experiments further validate the hypothesis.

**Claims And Evidence:**

Yes. both the issues claimed have supporting evidence in the form of ample experiments and analysis.

**Essential References Not Discussed:**

Not an expert in the field, so not sure.

**Ethical Review Concerns:**

None.

**Experimental Designs Or Analyses:**

Yes. the paper targets improvement in evaluation datasets MMLU and HellaSwag, so the use of them and the resulting improvements are totally valid.

**Methods And Evaluation Criteria:**

Yes. the paper targets improvement in evaluation datasets MMLU and HellaSwag, so the use of them and the resulting improvements are totally valid.

**Other Comments Or Suggestions:**

Please refer above.

**Other Strengths And Weaknesses:**

*Strengths*

- The paper addresses a very important problem of making sense of billion-scale training data for LLMs, by suggesting to group them into domains and styles along with an effective mixture technique.
- The method is easily scalable to bigger datasets, and the choice to run experiments with smaller models and extrapolate observations to bigger models is effective.
- The gains are complementary to quality metrics, so there is room for further improvement.

*Weaknesses*

- The whole technique to extract the mixture using RegMix is directly derived from a prior work RegMix, so the new ideas in that part are limited.
- The paper only focuses on improving two metrics, and scalability to improve multiple evaluation metrics seems unclear from the paper.

**Questions For Authors:**

Please refer to the weaknesses section above and answer the questions.

**Relation To Broader Scientific Literature:**

The paper provides a thorough comparison with the existing literature. It also includes comparison and contrast with respect to the quality filter literature, making it comprehensively clear the advantages in the paper.

**Theoretical Claims:**

No theoretical claims.

---

> ### Author Rebuttal · Authors · 2025-04-01
>
> Thank you for your review of our paper! We are glad that you highlighted our extensive experiments and analyses, as well as the scalability and practical value of our proposed approach. We would like to briefly address two of your points.
>
> > Reliance on RegMix
>
> In the paper, we clearly acknowledge that we use RegMix as an existing technique, since our focus is not on contributing another method for optimizing domain ratios given a set of data sources. Instead, we believe that the greater novelty of our work is to shift attention to defining the subsets for effective and large-scale data mixing. Nevertheless, we still extend RegMix to a novel setting, where we directly optimize the task loss of two downstream benchmarks (the bits-per-byte of the correct answers), instead of the perplexity on a corpus such as C4. In Appendix C, we also discuss several new modifications to RegMix that facilitate this, including fitting multiple per-task regression models and employing an iterative search method.
>
> > Scalability to improve multiple evaluation metrics is unclear
>
> We will make predictions of the topic and format mixtures for additional target tasks (the existing RegMix checkpoint only need to be evaluated on new tasks) and include these in the appendix in the final version of the paper. In our current draft, we focus on only two target tasks, MMLU and HellaSwag, since (a) these tasks are well known in the community and are commonly used for evaluating data selection due to their relevance and low variance (b) we can more easily associate characteristics of the combined mixture with one of the two tasks (c) it was not clear to us how many tasks should be averaged (e.g., should we assign a higher weight to MMLU than one of multiple commonsense tasks?) (d) we reserve more tasks to be used for evaluating generalization to “unseen” tasks. In fact, we find that mixtures for MMLU and HellaSwag generalize well across many popular benchmark tasks. However, we agree with the reviewer that finding the “right” target tasks will remain a question for future research.

---

### Official Review · Reviewer_FJKu · 2025-03-23

**Overall Recommendation:** 4

**Summary:**

The authors propose a framework for organizing domains from unstructured web corpora along dimensions of topic and format. They use a human-in-the-loop procedure to design domain taxonomies and train a small language model to automatically categorize documents in a pre-training corpus, a 200B token subset of CommonCrawl. Empirical experiments show that using their proposed notions of domain along both topic and format helps improve downstream task performance when combined with quality filters and domain reweighting, though reweighting data domains alone is shown not to match the benefits of filtering alone.

**Claims And Evidence:**

Claims supported

**Essential References Not Discussed:**

See other Comments for additional reference suggestions -- I would consider them relevant but not "essential"

**Experimental Designs Or Analyses:**

Sound experimentation

**Methods And Evaluation Criteria:**

Yes

**Other Comments Or Suggestions:**

Clarity issues and typos:
1. 2.2, "fine-tuning *a* small classifier model
2. I found the choice of wording in line 300 confusing ("Domain mixing is broadly effective" paragraph in 4.4). Unsure what exactly is meant by the "Despite the target task accuracy" sentence. In any case it does not seem essential to the paragraph, but I wonder if it might be more clear if the authors framed it as tasks being in-domain vs out-of-domain (if a matter of which tasks the domain mixture was geared towards), or simply mentioned the lower performance on HellaSwag specifically (if a matter of the underperformance on that single task)
3. Clarify what the "random sampling baseline" is referring to in 4.4's "Domain Mixtures improve quality filters" paragraph
4. I had trouble following the "Approximating quality filters by domains" paragraph in general, partially due to the density of typos (see below). More detail in the associated Table 2 caption could help. It is difficult to understand exactly what the "Baseline corpus" is out of context, and why perplexity score is expected to be lowest on it (see Q6 below). Explicitly framing the experiment as an ablation could help prevent misunderstandings as well: "document-level quality filtering is a far stronger intervention on the pre-training distribution than rebalancing domains or topics" feels possibly irrelevant to the key points of the paper. I would rather see a discussion of e.g. expected % values in Table 2 given particular hypotheses, or some kind of counterfactual reasoning about the results
5. Line 365, "In this case, *the* implicit Topic x Format mixture"
6. Line 370, "this classifier could *be* better suited"
7. Line 372, "values for domain mixing *are* substantially lower

Additional related work the authors may be interested in citing
1. [Chronopoulou et al., 2022](https://aclanthology.org/2022.naacl-main.96/) and [Reid et al., 2022](https://aclanthology.org/2022.emnlp-main.63/) propose two different hierarchical domain taxonomies. [Chronopoulou et al., 2022](https://aclanthology.org/2022.naacl-main.96/) propose a provenance-based taxonomy where leaf domains are from the top 100 URL domains from C4, and resulting domains may span both topic and format. [Reid et al., 2022](https://aclanthology.org/2022.emnlp-main.63/) propose two hierarchical levels of fine-grained and higher level topical domains. Relevant since the authors mention potential future work further breaking down their proposed domains into hierarchical taxonomies.
2. [Na et al., 2024](https://aclanthology.org/2024.emnlp-main.1176/) is another data mixing methodology paper, though "optimizing" data mixtures is not proposed as explicitly (no proportion reweighting experiments, only speculation of the method's extensibility towards reweighting). Relevant since the current submission's authors mention efficiency concerns from selecting a training mixture over the entire cross product of topic x format domains proposed and Na et al. propose a way to efficiently approximate fine-grained data ablations over many possible data mixtures with model merging instead of additional training. At the very least probably a Related Work citation (probably either Data mixture optimization or Analysis of pre-training data)
3. [Held et al., 2025](https://arxiv.org/abs/2501.11747) is very recent and only a pre-print but also feels pretty relevant as another data mixing methodology paper. I am not familiar enough with RegMix or UtiliMax to know whether UtiliMax is expected to be a promising alternative over RegMix for the current submission and/or any follow-up experiments, but I felt it is at the very least a related works citation (Data mixture optimization)
4. More influence functions literature for "Analysis of pre-training data": [Choe et al., 2024](https://arxiv.org/abs/2405.13954), [Yu et al., 2024](https://proceedings.neurips.cc/paper_files/paper/2024/hash/c4bec0d2fd217e6c2c3eafeced432582-Abstract-Conference.html)

I do not consider any of these to be "essential" references, necessarily, but I do feel engagement with these works could improve the overall quality of the paper

**Other Strengths And Weaknesses:**

Strengths:
1. Well-written -- a sincere pleasure to read overall
2. Strong empirical results
3. Compelling proposed domain taxonomy that may be useful to a broad audience
4. Interesting explorations across a broad set of ablations, including a k-means clustering baseline and a framing of quality filters as implicit domain reweighting
5. Overall high quality reproducibility efforts for established dataset filtering

Weaknesses:
1. Some clarity issues. In particular it is at times difficult to follow the reasoning for including the Section 5 experiment in the paper
2. Domain mixing results for only two downstream tasks are included, MMLU and HellaSwag (+ their combination), though the respective mixtures found are evaluated on a larger set of downstream tasks. I feel the paper would be stronger with more than just two domain mixing experiments, though I do not feel the current claims are necessarily undersupported as is. A skeptical reader may feel the results veer towards anecdotal
3. I would have liked to see more explicit discussion of limitations of the present study

**Questions For Authors:**

1. Is the proposed taxonomy expected to be applicable out of the box to more general LLM corpora (including e.g. synthetic data, proprietary data, and other content not typically found readily on the web)? If there are limitations, how feasible or compute intensive would it be to adapt the proposed taxonomy to fit a new web corpus e.g. adding just one more topical domain or one more format domain?
2. In general, I am curious to know what the work for 2.2 (Training domain classifiers for scaling annotations) and the related appendix section required in terms of GPU hours, and in particular how it compares to the projected cost of annotating documents with Llama 405b
3. Why was `gte-base-en-v1.5` chosen to compute document embeddings (as opposed to e.g. tf-idf embeddings used by [Gururangan et al](https://arxiv.org/abs/2303.14177), whom the authors cite as an exemplary baseline for k-means clustering of documents?
4. I had trouble following a detail of Target Tasks in Section 3: "To avoid contamination, we use the training and validation set of [MMLU and HellaSwag], respectively." How is this expected to help avoid contamination?
5. What are the key takeaways of Figure 4?
6. Are perplexity scores reported in Table 2 with respect to the baseline corpus's held-out data, or is there a different held-out set for each row?

**Relation To Broader Scientific Literature:**

Well-grounded in broader scientific literature. Very solid contribution, substantial enough but not groundbreaking to the point of being controversial

**Theoretical Claims:**

N/A

---

> ### Author Rebuttal · Authors · 2025-04-01
>
> Thank you for your detailed review and for regarding our paper as a solid contribution to the field with strong empirical results. We are especially grateful for the detailed suggestions with respect to the typos and clarity in Section 5, and the many interesting and relevant citations. We will improve the paper with your feedback and discuss these references in the related works section. We will also use the extra page of the final version to add a clear limitations section before the impact statement!
>
> > Domain mixing results for only two downstream tasks
>
> We focus on MMLU and HellaSwag, since these tasks exhibit interesting qualitative differences, are widely known in the community and are used by most prior works as stable and relevant metrics for data selection. However, we will include topic and format predictions for additional tasks in the appendix of the next version. Training additional models on many target task mixtures is unfortunately beyond our current computational resources and we believe that is equally important to hold out a set of tasks for measuring generalization to “unseen” tasks.
>
> Questions:
>
> 1. Applicable to more general LLM corpora?
>
> While we specifically designed our taxonomy with the goal of structuring many terabytes of web data, our domain classifiers can be applied to any text domain (we will also release a classifier that does not require an URL) and we would expect our models to produce reasonable results. Adapting the taxonomy is more complicated and computationally expensive — one would need to annotate additional data and retrain the domain classifiers — but our open-sourced scripts should make these operations run smoothly!
>
> 2. GPU time for training domain classifier?
>
> Great question! Fine-tuning the domain classifier is relatively cheap. The first stage of training takes less than 8 H100 hours and the second stage less than 1 H100 hour. By far the most extensive stage is annotating 100k web pages with Llama-3.1-405B, which takes approximately 600 H100 hours.
>
> 3. Why gte-base-en-v1.5?
>
> gte-base-en-v1.5 is the best small-scale embedding model on MTEB at the time of writing with support for longer documents (up to 8192 tokens). Unlike bag-of-word representations (like tf-idf), the neural encoder can produce embeddings which also reflect semantic and conceptual aspects of the source document.
>
> 4. Avoiding contamination
>
> The task examples for fitting RegMix and evaluating our final models are from completely disjoint sets, and we never use final test examples for fitting domains. Since our domain mixtures are very low-dimensional, it is unlikely to leak substantial information from the test sets, but it is still best practice to use a different split for mixture optimization.
>
> 5. Takeaways from Figure 4
>
> DCLM-fasttext and FineWeb-Edu are currently used to create some of the most widely used pre-training datasets and Figure 4 reflects the composition of these datasets. By comparing to the RegMix predictions, it also visualizes how the pre-training distributions change in the final four rows of Table 1, giving us some indication of which types of content are either “overlooked” or “overvalued” by the quality classifier. For example, we observe that FinWeb-edu has a strong inherent topic bias for Science, Health, and History, but overly filters  out more diverse topics. Meanwhile, DCLM-fasttext uniquely has strong preferences for the formats Comment Sections and Creative Writing. We will expand on this discussion in the next version of the paper.
>
>
> 6. Perplexity scores in Table 2
>
> These are computed on the same held-out data of the baseline corpus! We will add this for clarification.

---

> > ### Comment · Reviewer_FJKu · 2025-04-04
> >
> > Thank you for the detailed response to my review! In general, **I happily reaffirm my score of 4, as I feel strongly that this paper should be accepted.** Responding to some more specific points below:
> > - I do feel domain mixing for tasks beyond just MMLU and HellaSwag would strengthen the paper, so I look forward to seeing additional results in the appendix of the camera ready. However, I find the central claims satisfactory overall because I personally view the core contributions of the paper as those more closely tied to the proposed domain taxonomy. The domain mixing result is nice empirical evidence that the proposed domain taxonomy is a good and useful one that can help towards increasing performance on downstream tasks, but in my opinion there is utility in the inherent characteristics and construction of the taxonomy itself.
> > - It seems to me that the ordeal of adapting the existing proposed domain taxonomy to include new domains would be worth the compute costs to some parties. On the other hand, many would deem it infeasible or too expensive to redo the authors' annotation with Llama 405b themselves (and therefore appreciate the open release of the smaller trained annotation model). (I forgot to ask initially, but I think many would find it useful to understand the costs of performing inference with the smaller annotation model over the corpus as well). I encourage the authors to include these details of compute requirements in their revisions.
> >
> > I look forward to seeing the answers to my questions and concerns incorporated into the revisions.
> >
> > Additionally, I read the other reviews and rebuttals, and I found none of the other concerns to be major enough to influence my score downwards. In particular, the essential related work mentioned by Reviewer suAs seems to be a very recent, lesser known pre-print that has not yet passed peer review, so I do not believe it is [appropriate](https://icml.cc/Conferences/2025/ReviewerInstructions) to hold it against the authors of the present work for missing it (though I would encourage them to engage with it in any future revisions).

---

### Decision · Program_Chairs · 2025-05-01

**Decision:**

Accept (poster)

**Comment:**

This paper proposes a data curation method that organize the web corpora by developing taxonomies of their contents and organize them into domains. The allows the study of mixing data to improve models on downstream tasks. The work is novel and solid, and the results are promising. One remaining concern is the compute cost of the optimization of domain mixtures, which is higher than the actual DCLM 1B-1x pretraining that needs only 240 H100 hours. This point needs to be clarified and discussed. Also, the related work [Bai, Tianyi, etal] needs further discussion. Overall, I recommend acceptance given the solid contribution.